# How Data Mixing Shapes In-Context Learning: Asymptotic Equivalence for Transformers with MLPs

**Samet Demir[1], Zafer Doğan[1,2]***

[1]MLIP Research Group, KUIS AI Center, Koç University  [2]Department of EEE, Koç University
{sdemir20,zdogan}@ku.edu.tr

## Abstract

Pretrained Transformers demonstrate remarkable in-context learning (ICL) capabilities, enabling them to adapt to new tasks from demonstrations without parameter updates. However, theoretical studies often rely on simplified architectures (e.g., omitting MLPs), plain data models (e.g., linear regression with isotropic inputs), and single-source training—limiting their relevance to realistic settings. In this work, we study ICL in pretrained Transformers with nonlinear MLP heads on nonlinear tasks drawn from multiple data sources with heterogeneous input, task, and noise distributions. We analyze a model where the MLP comprises two layers, with the first layer trained via a single gradient step and the second layer fully optimized. Under high-dimensional asymptotics, we prove that such models are equivalent in ICL error to structured polynomial predictors, leveraging results from the theory of Gaussian universality and orthogonal polynomials. This equivalence reveals that nonlinear MLPs meaningfully enhance ICL performance—particularly on nonlinear tasks—compared to linear baselines. It also enables a precise analysis of data mixing effects: we identify key properties of high-quality data sources (low noise, structured covariances) and show that feature learning emerges only when the task covariance exhibits sufficient structure. These results are validated empirically across various activation functions, model sizes, and data distributions. Finally, we experiment with a real-world scenario involving multilingual sentiment analysis where each language is treated as a different source. Our experimental results for this case exemplify how our findings extend to real-world cases. Overall, our work advances the theoretical foundations of ICL in Transformers and provides actionable insight into the role of architecture and data in ICL.

## 1   Introduction

Transformers [38] have emerged as a dominant architecture across a broad range of machine learning applications. A particularly striking capability of these models is *in-context learning* (ICL)—the ability to adapt to new tasks from a few examples presented as input, without parameter updates [7]. As this phenomenon becomes central to modern AI systems, a deeper theoretical understanding of when and why Transformers succeed at ICL is increasingly critical.

To make this challenge tractable, recent work has studied simplified settings, often focusing on linear tasks and attention-only models that omit multilayer perceptrons (MLPs) [3, 42, 45, 47]. However, such abstractions overlook the empirical observation that nonlinear MLPs are integral to Transformer performance in practice. While a few recent studies have begun to investigate MLPs in the ICL context [23, 24, 32], their analyses are restricted either to nonstandard architectures, specific activation types, or overly simplistic data settings—typically assuming a single, homogeneous data source.

---

*Corresponding author
Source code: https://github.com/KU-MLIP/Data-Mixing-Shapes-ICL-by-Transformers.

39th Conference on Neural Information Processing Systems (NeurIPS 2025).

In contrast, real-world settings involve Transformers with nonlinear MLPs trained with data from multiple heterogeneous sources, where data quality vary. This raises two open questions:
*(i) How do nonlinear MLPs shape ICL behavior under realistic data conditions? (ii) What role does mixture of training data play in shaping the Transformers' capabilities for ICL and feature learning?*

To address these questions, we analyze the ICL performance of Transformers with a nonlinear MLP head trained under multiple data sources with different distributions, focusing on nonlinear regression tasks in high-dimensional regimes. We adopt an asymptotic lens where sample size, context length, and hidden dimension jointly diverge, and draw on Gaussian universality theory [21, 13] to analytically characterize model behavior. Specifically, we study a Transformer with linear attention and a two-layer nonlinear MLP head, where the first layer is trained with a single gradient step and the second is fully trained. We show that this model is asymptotically equivalent to a finite-degree polynomial model in terms of ICL error. This equivalence allows us to (i) explain the performance gains due to nonlinear MLPs and (ii) study how different training data distributions affect ICL outcomes.

Our theoretical findings are corroborated by simulations, which demonstrate that nonlinear MLPs significantly improve ICL performance over linear counterparts. We further reveal that data sources with structured covariances and low noise are critical for effective learning. Next, we show that feature learning in Transformers depends crucially on the structure of task distributions, and that certain mixtures can either enhance or suppress the model's ability to learn useful features. Finally, we provide an experimental result on a real-world scenario (multilingual sentiment analysis), displaying how our results can apply in a scenario where different languages are treated as different sources.

Overall, our contributions are as follows:

1. We show that a Transformer with a nonlinear MLP head is asymptotically equivalent to a polynomial model, and outperforms its linear counterpart on nonlinear ICL tasks.
2. We analyze the role of data mixing in ICL and identify key properties—structured covariances for input and task vectors and low target noise—that define high-quality data sources.
3. We uncover the interaction between data mixing and feature learning, showing that structure in task distributions is essential for meaningful feature learning to occur.

## 2   Related Work

**In-context learning with Transformers**   The discovery of in-context learning (ICL) in large-scale Transformers [7] has prompted a surge of empirical and theoretical investigations. Empirically, studies such as [44, 33, 37] highlight that ICL capabilities grow with model size, suggesting an emergent behavior critical to modern foundation models. On the theoretical side, controlled synthetic tasks—especially linear regression—have become a standard benchmark for analyzing ICL mechanisms in simplified settings [47, 16, 36]. These studies often adopt linearized attention mechanisms for tractability and argue that Transformers learn to simulate algorithmic procedures during pre-training [5, 3, 2, 25, 42, 29, 15, 26, 35]. Yet, the precise nature of these learned procedures remains unclear. Notably, [47, 27, 28] offer generalization analysis for Transformers with a linear attention (without MLPs) in linear ICL tasks, laying the groundwork for extensions to more realistic settings.

**Nonlinear MLPs in Transformers**   While much of the theoretical literature abstracts away the MLP components of Transformers, recent efforts have started to examine their impact on ICL. For example, [24] studied ReLU-based MLPs in classification tasks, while [23, 32] analyzed architectures where MLP layers precede attention—deviating from the standard Transformer design in which MLPs follow attention blocks. Furthermore, [1] studied a simplified Transformer architecture with a fixed (non-trainable) nonlinear attention mechanism—omitting the training of key, query, and value weights—and an optional MLP head. Overall, these studies either restrict task types, limit activation functions, adopt nonstandard architectures, or omit training of the attention weights, leaving open questions about the role of nonlinear MLPs in Transformers for nonlinear ICL in realistic settings. Our work addresses this gap by analyzing a Transformer with a trained nonlinear MLP head consistent with the canonical architecture and showing asymptotic equivalence of a polynomial surrogate model to the Transformer model in terms of the ICL error. Moreover, prior works focus primarily on single-source training with simplified data distributions, whereas we consider multiple heterogeneous data sources and explicitly characterize the impact of data mixture ratios on the ICL performance.

**Gaussian universality** The concept of Gaussian universality from random matrix theory has proven valuable for the asymptotic analysis of neural networks. Under isotropic Gaussian inputs, random feature models (two-layer neural network with random first layer and trained second layer) can be shown to be equivalent to Gaussian models with matched first and second moments [17, 21, 8]. These results have been extended to broader settings, including empirical risk minimization [31], mixture-distributed inputs [11], and structured covariances [12]. More recently, the equivalence has been rigorously established for two-layer networks trained with one gradient step [30, 10, 13]. However, these universality results have not yet been extended to Transformer architectures or in-context learning scenarios. Our work bridges this gap by showing that a Transformer with a nonlinear MLP is asymptotically equivalent to a finite-degree polynomial model—thus connecting Gaussian universality theory to Transformer-based ICL for complex data scenarios involving data mixing.

## 3 Problem formulation

### 3.1 In-context learning of nonlinear regression

We investigate the in-context learning (ICL) capabilities of pretrained Transformer architectures on nonlinear regression tasks. Given a sequence of input-output pairs—referred to as the *context*—

$$\big(\boldsymbol{x}_1, y_1\big),\ \big(\boldsymbol{x}_2, y_2\big), \ldots, \big(\boldsymbol{x}_\ell, y_\ell\big),\ \big(\boldsymbol{x}_{\ell+1}, ?\big),$$

where each input $\boldsymbol{x}_i \in \mathbb{R}^d$ and corresponding response $y_i \in \mathbb{R}$ are sampled i.i.d. from an unknown joint distribution, the objective is to predict $y_{\ell+1}$ for a new input $\boldsymbol{x}_{\ell+1}$ using the context. The context length is denoted by $\ell$. In the nonlinear setting, we assume the input-output relationship cannot be captured by any linear function. Thus, the model has to learn to estimate the nonlinear input-output relationship from a set of training contexts.

### 3.2 Data model

While most of the theoretical works assume a single data source, in practice, the models are pretrained with datasets consisting of a mixture of data sources [46]. To reflect this, we consider $\mathcal{S}$ different data sources. Thus, when constructing a dataset, we first sample the source index $s$ from a categorical distribution with the following probabilities: $\mathbb{P}(s = i) = \rho_i$ for $\{0, 1, \ldots, \mathcal{S}-1\}$. Then, for each data source, we posit a nonlinear relationship between $\boldsymbol{x}$ and $y$ governed by a context-specific parameter vector $\boldsymbol{\xi} \in \mathbb{R}^d$, subject to additive Gaussian noise. Although $\boldsymbol{\xi}$ remains constant within a given context, it is re-sampled across different contexts, imposing the requirement that the model estimate $\boldsymbol{\xi}$ from the observed pairs before generalizing to the new input.

Formally, for a given source index $s$, we first sample the task vector $\boldsymbol{\xi}|s \sim \mathcal{N}(\boldsymbol{\mu}_{\xi,s}, \boldsymbol{\Sigma}_{\xi,s})$ and using the sampled task vector, data is sampled as:

$$\boldsymbol{x}_i|s \sim \mathcal{N}(\boldsymbol{\mu}_{x,s}, \boldsymbol{\Sigma}_{x,s}),\ \text{and}\ y_i|s := \phi_s\left(\frac{(\boldsymbol{\xi}|s)^T(\boldsymbol{x}_i|s)}{\|\boldsymbol{\xi}|s\|_2 \|\boldsymbol{\Sigma}_{x,s}\|_2^{1/2}}\right) + \epsilon_i|s, \tag{1}$$

where $\phi_s : \mathbb{R} \to \mathbb{R}$ is a nonlinear function and $\epsilon_i|s \sim \mathcal{N}(0, \Delta_s^2)$ is noise. This setup encapsulates structured nonlinear regression and classification tasks with mixed data sources.

Following [47], we construct an embedding matrix $\boldsymbol{Z} \in \mathbb{R}^{(d+1)\times(\ell+1)}$ by stacking the feature vectors and their corresponding labels, using a zero placeholder for the unknown label:

$$\boldsymbol{Z} = \begin{bmatrix} \boldsymbol{x}_1|s & \boldsymbol{x}_2|s & \cdots & \boldsymbol{x}_\ell|s & \boldsymbol{x}_{\ell+1}|s \\ y_1|s & y_2|s & \cdots & y_\ell|s & 0 \end{bmatrix} \in \mathbb{R}^{(d+1)\times(\ell+1)}, \tag{2}$$

where we omit the source index $s$ for $\boldsymbol{Z}$ to streamline the presentation.

### 3.3 Transformer model

In this work, we consider a Transformer model involving single block / layer of a linear attention and an MLP head (two-layer neural network). While it is possible to extend the setting to those involving multiple blocks of attention and MLP layers, following many theoretical studies [47, 27, 3] in the literature, we focus on the single block case to streamline our theoretical analysis. Similarly, while

nonlinear attention mechanisms (such as softmax attention) are popular in the empirical literature, we use linear attention since it is analytically tractable [47, 27] and variants of linear attention can lead to performance similar to softmax attention [20]. Another benefit of focusing on linear attention is that it allows us to isolate the role of the nonlinear MLP head in capturing the nonlinear nature of our ICL tasks since nonlinear attention can also play a role in capturing such nonlinear relations. In the rest of this section, we first describe the linear attention mechanism–introducing linear Transformer without MLP head as our baseline– and then, we explain our Transformer model with MLP head together with the training procedure.

### 3.3.1 Linear attention (linear Transformer)

The Transformer must leverage this representation to infer the underlying mapping and predict $y_{\ell+1}$. Then, the output of linear attention in the Transformer can be computed as:

$$A := Z + \frac{1}{\ell} V Z (KZ)^T (QZ), \tag{3}$$

where $K, Q, V$ are appropriately sized key, query, and value matrices, respectively. When predicting $y_{\ell+1}$, the relevant output of the linear Transformer (without MLPs) will be $A_{d+1,\ell+1}$, i.e., the element of $A$ corresponding to the 0 element in the embedding matrix $Z$. Using the reparameterization in [27] (see Appendix A), this prediction can be simplified to:

$$\hat{y}_{\text{linear}} = \text{vec}(\mathbf{\Gamma})^T \text{vec}(H_Z), \tag{4}$$

where $\text{vec}(.)$ denotes the vectorization operation, $\mathbf{\Gamma} \in \mathbb{R}^{d \times (d+1)}$ is the parameter matrix formed using the entries of $K, Q, V$ matrices while $H_Z$ is defined as

$$H_Z := x_{\ell+1} \left[ \frac{1}{\ell} \sum_{i \leq \ell} y_i x_i^T \quad \frac{1}{\ell} \sum_{i \leq \ell} y_i^2 \right] \in \mathbb{R}^{d \times (d+1)}.$$

Note that the parameter matrix $\mathbf{\Gamma}$ is trained for the linear Transformer case as

$$\arg\min_{\mathbf{\Gamma}} \frac{1}{n} \sum_{j=1}^n \left( y_{\ell+1}^j - \text{vec}(\mathbf{\Gamma})^T \text{vec}(H_{Z^j}) \right)^2 + \lambda \|\mathbf{\Gamma}\|_F^2,$$

where $\lambda$ is the regularization constant. Here, $\{(Z^j, y_{\ell+1}^j)\}_{j=1}^n$ denotes the $n$ samples (contexts) used for the training where $Z^j$ is formed by (2).

### 3.3.2 Transformer with a nonlinear MLP

ICL of nonlinear tasks requires nonlinear models. As such, Transformers include nonlinear MLP layers for capturing such nonlinearities. Therefore, we consider the Transformer with a nonlinear MLP head, for which the prediction of the model can be written as

$$\hat{y}_{\text{nonlinear}} := \frac{1}{\sqrt{k}} w^T \sigma(F \text{vec}(H_Z)), \tag{5}$$

where $\sigma : \mathbb{R} \to \mathbb{R}$ is the activation of MLP, and the parameters are $F \in \mathbb{R}^{k \times d(d+1)}$ and $w \in \mathbb{R}^k$. Here, $F$ encapsulates the parameters of the attention (denoted by $\mathbf{\Gamma}$ in the description of the linear attention above) and the parameters of the first layer in the MLP head while $w$ is the weights corresponding to the second layer of the MLP.

**Training procedure** To simplify our analysis while keeping feature learning for the MLP, we consider the following training procedure with two stages [4, 9]:

**i) Gradient descent on the first layer:** Given $\{(\tilde{Z}^j, \tilde{y}_{\ell+1}^j)\}_{j=1}^n$ a set of training samples drawn using the procedure described above, we first fix $w$ at the initialization and perform a single gradient descent step on $F$ with respect to squared loss. The gradient update with *step size $\eta > 0$* is given as

$$\hat{F} := F + \eta G, \tag{6}$$

where gradient matrix $G$ is defined as

$$G := \frac{1}{n} \left( \frac{1}{\sqrt{k}} \left( w \tilde{y}^T - \frac{1}{\sqrt{k}} w w^T \sigma(F \tilde{H}^T) \right) \odot \sigma'(F \tilde{H}^T) \right) \tilde{H}, \tag{7}$$

with $\tilde{H} := [\text{vec}(H_{\tilde{Z}^1}), \text{vec}(H_{\tilde{Z}^2}), \dots, \text{vec}(H_{\tilde{Z}^n})]^T$, and $\tilde{y} := [\tilde{y}_{\ell+1}^1, \tilde{y}_{\ell+1}^2, \dots, \tilde{y}_{\ell+1}^n]^T$.

**ii) Ridge regression for the second layer:** A fresh sample set $\{(\boldsymbol{Z}^j, y_{\ell+1}^j)\}_{j=1}^n$ is used to train $\boldsymbol{w}$:

$$\hat{\boldsymbol{w}} := \arg\min_{\boldsymbol{w}} \frac{1}{n} \sum_{j=1}^{n} \left( y_{\ell+1}^j - \frac{1}{\sqrt{k}} \boldsymbol{w}^T \sigma(\hat{\boldsymbol{F}} \text{vec}(\boldsymbol{H}_{\boldsymbol{Z}^j})) \right)^2 + \lambda \|\boldsymbol{w}\|_2^2,$$

where $\lambda \geq 0$ is the regularization constant.

Here, by avoiding data reuse, we ensure $\hat{\boldsymbol{F}}$ remains independent of the second training set, which simplifies theoretical analysis [4]. Furthermore, while our training procedure differs from standard end-to-end training techniques, this two-phase scheme –gradient descent on the first layer followed by ridge regression on the second– is well-motivated by prior theoretical work on MLPs [4, 9]. This separation enables tractable analysis while preserving meaningful feature learning dynamics.

### 3.4 Evaluating ICL Performance

To evaluate ICL performance, we consider a uniform mixture of data sources, i.e., $\mathbb{P}(s = i) = 1/\mathcal{S}$ for all $i \in \{0, 1, \dots, \mathcal{S} - 1\}$. Under this setting, we define the ICL error as

$$\frac{1}{\mathcal{S}} \sum_{\hat{s}=0}^{\mathcal{S}-1} \mathbb{E}\left[ (y_{\ell+1} - \hat{y})^2 \,\big|\, s = \hat{s} \right], \tag{8}$$

where $\hat{y}$ denotes the model prediction, given by (4) for the linear Transformer or by (5) for the Transformer with a nonlinear MLP. This formulation enables controlled experimentation with training-time data mixtures while evaluating performance as an average across all sources. Such an approach aligns with practical multi-source training scenarios, where models are trained on heterogeneous datasets and validated by averaging performance over the constituent sources [46]. Note that, we also provide ICL errors corresponding to each source (for cases when it might be relevant) in Appendix G.

## 4 Main results

In this section, we present both our theoretical and empirical results. We begin by outlining the assumptions underlying our asymptotic analysis. Next, we introduce our main theoretical result establishing an asymptotic equivalence. We conclude with simulation studies and a real-world experiment that validate this equivalence and further illustrate how performance is influenced by factors such as sample size, context length, hidden dimension, feature learning, and data mixing.

### 4.1 Assumptions

We formalize our theoretical analysis under a set of assumptions designed to capture the high-dimensional nature of in-context learning (ICL) and the architectural properties of Transformers.

**Assumption 4.1** (Proportional limit). The input dimension $d$, context length $\ell$, number of training samples $n$, and hidden dimension $k$ diverge jointly as $d, \ell, n, k \to \infty$, while maintaining constant ratios $\ell/d$, $n/d^2$, and $k/n$ in $\mathbb{R}^+$.

This assumption aligns with regimes studied in prior work: $\ell/d$ and $n/d^2$ are critical for ICL performance in linear Transformers [27], while $k/n$ ensures the model capacity scales with data [21].

**Assumption 4.2** (Input statistics). The covariance matrices of input vectors satisfy $\|\boldsymbol{\Sigma}_{x,s}\|_2^2 = \mathcal{O}(d)$ and $\text{Tr}(\boldsymbol{\Sigma}_{x,s}) \asymp d$ while the mean vectors satisfy $\|\boldsymbol{\mu}_{x,s}\|_2^2 = \mathcal{O}(d^{1/2})$.

The assumption on the input statistics naturally constrains the input distribution. To streamline the exposition, we restrict our analysis to the zero-mean case ($\boldsymbol{\mu}_{x,s} = \boldsymbol{0}$), with the generalization to non-zero mean inputs discussed in Appendix F.

**Assumption 4.3** (Scaling of the step size). The step size $\eta$ scales as $\eta = o(d^2)$.

By permitting step size to scale with the input dimension while the dimension tends to infinity, we introduce nontrivial feature learning, allowing the model to learn nonlinear features [4, 13].

**Assumption 4.4** (Covariance of the attention output). For all task indices $i, j$, the attention output satisfies
$$\text{Tr}(\text{Cov}(\text{vec}(\boldsymbol{H_Z})|s = i)) = \text{Tr}(\text{Cov}(\text{vec}(\boldsymbol{H_Z})|s = j)),$$
and admits a low-rank decomposition:
$$\text{Cov}(\text{vec}(\boldsymbol{H_Z})|s) = \boldsymbol{I}_{d(d+1)} + \sum_{q=1}^{r_s} \theta_{s,q} \boldsymbol{\gamma}_{s,q} \boldsymbol{\gamma}_{s,q}^T, \tag{9}$$
where $r_s \in \mathbb{Z}^+$, $\theta_{s,q} > 0$, and $\{\boldsymbol{\gamma}_{s,q}\}$ are orthonormal vectors in $\mathbb{R}^{d(d+1)}$.

**Assumption 4.5** (Joint scaling). The step size and covariance of attention outputs satisfy
$$\eta \cdot \|\text{Cov}(\text{vec}(\boldsymbol{H_Z}))\|_2 = o(d^2).$$

Assumptions 4.3- 4.5 ensure tractability in our asymptotic analysis while allowing the feature learning. Although these assumptions might look unnatural at first glance, they enable us to connect our setting involving ICL by Transformer with a nonlinear MLP to prior results [13] for MLPs, facilitating our theoretical analysis. We would like to note that Assumptions 4.3- 4.5 represent limitations of our theoretical results. Nevertheless, our experimental results on both synthetic and real-world datasets suggest that these assumptions can be partially relaxed in practice.

**Assumption 4.6** (Initialization). The vector $\boldsymbol{w} \sim \mathcal{N}(0, \frac{\boldsymbol{I}_k}{k})$ and the matrix $\boldsymbol{F} := [\boldsymbol{f}_1, \ldots, \boldsymbol{f}_k]^T$ has rows $\boldsymbol{f}_i \sim \mathcal{N}\left(0, \frac{\boldsymbol{I}_{d(d+1)}}{\text{Tr}(\text{Cov}(\text{vec}(\boldsymbol{H_Z})))}\right)$.

This initialization is consistent with standard practices for MLP initializations [18].

**Assumption 4.7** (Target function). Each task-specific function $\phi_s : \mathbb{R} \to \mathbb{R}$ is Lipschitz.

**Assumption 4.8** (Activation function). The activation $\sigma : \mathbb{R} \to \mathbb{R}$ has bounded derivatives and satisfies $\mathbb{E}_{z \sim \mathcal{N}(0,1)}[\sigma(z)^2] < \infty$, admitting a Hermite expansion [34, Chapter 11.2]:
$$\sigma(x) = \sum_{j=0}^{\infty} \frac{1}{j!} h_j H_j(x), \quad h_j = \mathbb{E}_{z \sim \mathcal{N}(0,1)}[H_j(z)\sigma(z)], \tag{10}$$
where $H_j$ is the $j$-th probabilist's Hermite polynomial.

Assumptions 4.7–4.8 accommodate a broad class of nonlinearities for both the target and activation functions, while maintaining analytical tractability via Hermite expansions of the activation. Although certain functions, such as polynomials, may not strictly satisfy the bounded-derivative condition, our theoretical results are expected to remain valid when the derivatives of the activation are bounded with high probability for inputs $z \sim \mathcal{N}(0, 1)$.

## 4.2 Asymptotic equivalence

We present our theoretical results in terms of an asymptotic equivalence between models. The main technical challenge in our analysis arises from characterizing the distribution of the attention outputs, denoted by $\text{vec}(\boldsymbol{H_Z})$. To address this, we begin with an equivalent statistical representation of the random feature mapping $\boldsymbol{F}\text{vec}(\boldsymbol{H_Z})$, as formalized in the following lemma. This representation later serves as the foundation for our analysis of the Transformer with a nonlinear MLP (5).

**Lemma 4.9** (Asymptotic distribution of $\boldsymbol{F}\text{vec}(\boldsymbol{H_Z})$).
*Under Assumption 4.6, the entries of $\boldsymbol{F}$ are i.i.d. $\mathcal{N}(0, 1/\text{Tr}(\text{Cov}(\text{vec}(\boldsymbol{H_Z}))))$, ensuring that the components of $\boldsymbol{F}\text{vec}(\boldsymbol{H_Z})$ have unit variance. Let $\boldsymbol{f}_i$ denote the $i$-th row of $\boldsymbol{F}$. Given the assumptions stated in Section 4.1, we obtain*
$$\boldsymbol{f}_i^T \text{vec}(\boldsymbol{H_Z}) \to \mathcal{N}(0, 1) \text{ converges in distribution}, \tag{11}$$
*for all $i \in \{1, \ldots, k\}$.*

*Proof.* Set $t := \text{Tr}(\text{Cov}(\text{vec}(\boldsymbol{H_Z})))$. Given $\boldsymbol{H_Z}$, we have
$$\boldsymbol{f}_i^T \text{vec}(\boldsymbol{H_Z}) \mid \boldsymbol{H_Z} \sim \mathcal{N}(0, \|\text{vec}(\boldsymbol{H_Z})\|_2^2/t). \tag{12}$$
Since $\|\text{vec}(\boldsymbol{H_Z})\|_2^2/t$ concentrates around 1, the desired result follows. See Appendix B. □

The lemma simplifies the term $\boldsymbol{F}\text{vec}(\boldsymbol{H_Z})$, enabling a more tractable analysis for our study of in-context learning in Transformers with nonlinear MLPs. As a result, we concentrate on the joint behavior of $(\boldsymbol{F}\text{vec}(\boldsymbol{H_Z}), \boldsymbol{\xi}^T \boldsymbol{x}_{\ell+1})$ rather than $(\boldsymbol{H_Z}, y_{\ell+1})$, given that $y_{\ell+1} := \phi_s(\boldsymbol{\xi}^T \boldsymbol{x}_{\ell+1}) + \epsilon_{\ell+1}$. The subsequent corollary demonstrates that $(\boldsymbol{F}\text{vec}(\boldsymbol{H_Z}), \boldsymbol{\xi}^T \boldsymbol{x}_{\ell+1})$ converges in distribution to a jointly Gaussian vector in the high-dimensional limit that we consider.

**Corollary 4.10** (Joint distribution of $(\boldsymbol{F}\text{vec}(\boldsymbol{H_Z}), \boldsymbol{\xi}^T \boldsymbol{x}_{\ell+1})$ conditioned on $\boldsymbol{\xi}$ and $s$). *Suppose that the task vector $\boldsymbol{\xi}$ and data source $s$ are given. Under our assumptions (given in Section 4.1), $(\boldsymbol{F}\text{vec}(\boldsymbol{H_Z}), \boldsymbol{\xi}^T \boldsymbol{x}_{\ell+1})$ becomes jointly Gaussian with a certain mean and covariance.*

Lemma 4.9 and Corollary 4.10 collectively characterize the distributions of both the $\boldsymbol{F}\text{vec}(\boldsymbol{H_Z})$ and the label $y_{\ell+1}$, which are utilized for our subsequent study of the asymptotic equivalence.

Next, we need to consider the impact of updating $\boldsymbol{F}$ with one gradient descent step as described in (6): $\hat{\boldsymbol{F}} := \boldsymbol{F} + \eta \boldsymbol{G}$ where $\boldsymbol{G}$ is the gradient matrix and $\eta$ is the step size. For this purpose, we look into the gradient matrix $\boldsymbol{G}$ in the following lemma.

**Lemma 4.11** (Decomposition of the gradient matrix). *The gradient $\boldsymbol{G}$ defined in (7) admits the following decomposition: $\boldsymbol{G} = \boldsymbol{u}\boldsymbol{v}^T + \boldsymbol{\Delta}$, where $\boldsymbol{u} := \alpha \boldsymbol{w}$ for some $\alpha \in \mathbb{R}$ and $\boldsymbol{v} := \tilde{\boldsymbol{H}}^T \tilde{\boldsymbol{y}}/(m\sqrt{k})$. Here, $\boldsymbol{u}\boldsymbol{v}^T$ is the dominant rank-one term while $\|\boldsymbol{\Delta}\|_2$ is the negligible residual term.*

*Proof.* Using high-probability tail bounds, this can be shown similarly to the decompositions of the gradient matrix for simpler neural network settings in the literature [4, 13]. See Appendix C. □

With these components established, we can now leverage the preceding results in conjunction with existing asymptotic analyses of two-layer neural networks [13] to study the in-context learning error defined in (8) for Transformers equipped with nonlinear MLPs. The following theorem identifies a polynomial model that is asymptotically equivalent to such a Transformer architecture.

**Theorem 4.12** (Equivalent polynomial model). *Suppose our assumptions described in Section 4.1. Consider we train the first layer $\hat{\boldsymbol{F}}$ of the MLP according to (6). Then, the Transformer with a nonlinear MLP in (5) performs asymptotically equivalent to the model*

$$\frac{1}{\sqrt{k}} \boldsymbol{w}^T \hat{\sigma}_p(\hat{\boldsymbol{F}}\text{vec}(\boldsymbol{H_Z})) \tag{13}$$

*with respect to the ICL error (8), where the second layers $\boldsymbol{w}$ in the two models are trained separately. Here, $\hat{\sigma}_p : \mathbb{R} \to \mathbb{R}$ is a degree-$p$ polynomial function with a residual term, which is defined as*

$$\hat{\sigma}_p(x) := \sum_{i=0}^{p} \frac{1}{i!} c_i H_i(x) + c_p^* z \ \ \text{for} \ \ z \sim \mathcal{N}(0, 1), \tag{14}$$

*where $H_i : \mathbb{R} \to \mathbb{R}$ denotes the $i$-th (probabilist's) Hermite polynomial, $c_i$ are the corresponding Hermite coefficients and $c_p^*$ is the residual term such that $\mathbb{E}_{x \sim \mathcal{N}(0,1)}[\hat{\sigma}_p(x)^2] = \mathbb{E}_{x \sim \mathcal{N}(0,1)}[\sigma(x)^2]$. The smallest degree $p$ that is necessary for the equivalence depends on the joint distribution of $(\hat{\boldsymbol{F}}\text{vec}(\boldsymbol{H_Z}), \boldsymbol{\xi}^T \boldsymbol{x}_{\ell+1})$ and the activation function $\sigma$, but a finite $p$ suffices under our assumptions.*

*Proof.* We employ our established theoretical results 4.9-4.11–regarding the distribution of $(\boldsymbol{F}\text{vec}(\boldsymbol{H_Z}), \boldsymbol{\xi}^T \boldsymbol{x}_{\ell+1})$– in combination with the orthogonality property of Hermite polynomials under the Gaussian measure in order to prove this theorem. See Appendix E. □

Theorem 4.12 establishes an asymptotic equivalence between the ICL behavior of a Transformer with an MLP and that of a polynomial model in terms of the ICL errors. This result is significant for several reasons. Firstly, (13) admits a much simpler analysis, allowing precise characterization of ICL error behavior of the Transformer under data mixing and feature learning. Second, it sheds light on the function class learned by the Transformer—specifically, that it effectively learns a low-degree polynomial approximation of the task function. This equivalence opens the door to optimizing nonlinearities in MLPs (inside the Transformer) through polynomial surrogate analysis. These insights extend beyond classical intuition and align with recent trends in Gaussian equivalence literature [21, 30, 13]. Note that although similar equivalence results exist for supervised learning by two-layer neural networks [30, 13], this work represents a cornerstone in adapting those results to a novel setting involving ICL by Transformers with MLPs.

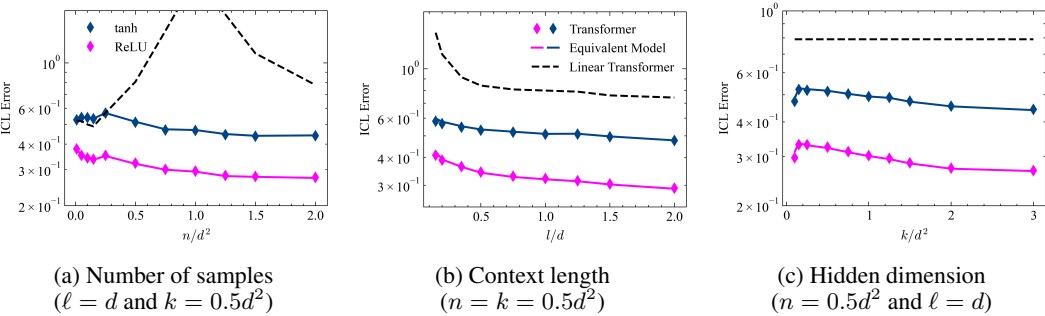

|  (a) Number of samples | (b) Context length | (c) Hidden dimension |
| $(\ell = d$ and $k = 0.5d^2)$ | $(n = k = 0.5d^2)$ | $(n = 0.5d^2$ and $\ell = d)$ |

Figure 1: Effects of sample size, context length, and hidden dimension on the ICL errors for linear Transformer (4), Transformer with a nonlinear MLP (5), and the equivalent model (13). The number of data sources is $\mathcal{S} = 2$ with equal probability: $\mathbb{P}(s = 0) = \mathbb{P}(s = 1) = 1/2$. For the input vectors, $\boldsymbol{\mu}_{x,s} = \mathbf{0}$ and $\boldsymbol{\Sigma}_{x,s} = \boldsymbol{I}_d$ for all $s$. For the task vectors, $\boldsymbol{\mu}_{\xi,s} = \mathbf{0}$ for all $s$ while $\boldsymbol{\Sigma}_{\xi,0} = \boldsymbol{I}_d$ and $\boldsymbol{\Sigma}_{\xi,1} = \boldsymbol{I}_d + \theta\boldsymbol{\gamma}\boldsymbol{\gamma}^T$ for some $\theta \asymp d^2$ and $\boldsymbol{\gamma} \in \mathbb{R}^d$ with $\|\boldsymbol{\gamma}\|_2 = 1$. The target function $\phi_s$ is ReLU, and the target noise is $\Delta_s = 0.01$ for all $s$. The Transformer is used with two different activation functions: ReLU and tanh. Here, $d = 80$, $\eta \asymp d^2$, and $\lambda = 5 \times 10^{-5}$. The degree of the equivalent polynomial model $p$ is set to $4$. The average over 20 Monte Carlo runs is plotted.

## 4.3 Experimental results

Leveraging our equivalence results, we systematically compare the performance of the Transformer with a nonlinear MLP to that of the linear Transformer. We also empirically validate the equivalence between the Transformer and its polynomial surrogate, highlighting their predictive alignment under various conditions. We illustrate the data mixing effects in ICL exhibited by Transformers with a nonlinear MLP. Next, we provide results depicting the interplay between feature learning, data mixing, and properties of the data sources. Finally, we consider an example real-world scenario involving multilingual sentiment analysis and display how our findings extend to this case.

### 4.3.1 Data setup

In our experiments, we consider a setting with two data sources. For the synthetic (simulation) scenarios, the first source is an isotropic Gaussian in both input and task space, representing a low-quality (noise-like) dataset, while the second source features non-isotropic covariance in either the input or task distributions, introducing additional structure and serving as a higher-quality signal. This configuration enables us to analyze how ICL performance varies under mixtures of differing data quality. The rest of the experimental details and the setting of our real-world (multilingual sentiment analysis) experiment are described in the corresponding captions.

### 4.3.2 Effect of model dimensions

We first assess the role of sample size ($n$), context length ($\ell$), and hidden dimension ($k$) on ICL error. Results are shown in Figure 1. We highlight two consistent observations across all cases:

(i) The nonlinear MLP Transformer (5) outperforms the linear one (4) in terms of ICL error (8). Although outperforming the linear Transformer may sound trivial, without the training of the first layer $\boldsymbol{F}$ as in (6), there exist many cases where the Transformer with the nonlinear MLP head fails to outperform the linear one [14]. Therefore, outperforming the linear one is a reasonable baseline and an indicator of effective feature learning.

(ii) The nonlinear MLP Transformer closely aligns with its polynomial surrogate model (13), even at moderate dimensionalities, empirically validating our asymptotic equivalence results.

These findings underscore both the benefits of nonlinear MLPs and the theoretical significance of our equivalence framework. Additionally, we observe a characteristic double descent phenomenon [6] in the ICL error: with respect to sample size in Figure 1(a) and hidden dimension in Figure 1(c). The effect of increasing context length $\ell$ is shown in Figure 1(b), revealing a consistent reduction in ICL error, reflecting improved task estimation with longer contexts.

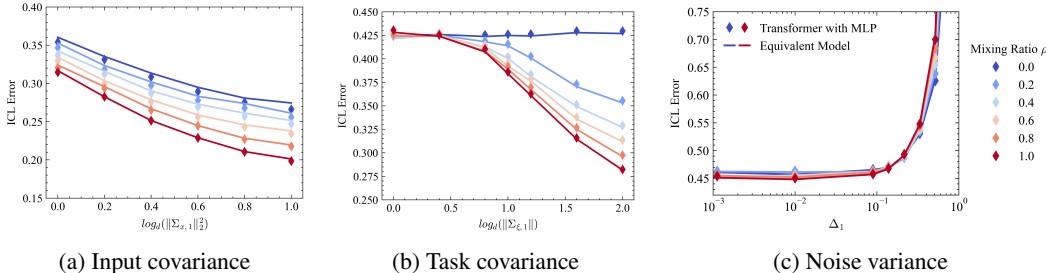

|        |                   |                 |                  |
|--------|-------------------|-----------------|------------------|
| (a) Input covariance | (b) Task covariance | (c) Noise variance |

Figure 2: Data mixing effects on the ICL error: performance of Transformers with ReLU activation (denoted with diamonds (5)), and the equivalent model (denoted with lines (13)) are illustrated for different mixing ratios $\rho$. For this figure, $\rho$ controls the data source mixture for the training, as we define $\mathbb{P}(s=0) := 1 - \rho$ and $\mathbb{P}(s=1) := \rho$, while the ICL error is calculated as an average over data sources (8). Here, $d = 80$, $l = d$, $n = k = 0.5d^2$, $\lambda = 5 \times 10^{-5}$, $\eta \asymp d^2$ and the target function $\phi_s$ is ReLU for all $s$. We initially consider the following setting: for the input vectors, $\boldsymbol{\mu}_{x,s} = \mathbf{0}$ and $\boldsymbol{\Sigma}_{x,s} = \boldsymbol{I}_d$ for all $s$; for the task vectors, $\boldsymbol{\mu}_{\xi,s} = \mathbf{0}$ and $\boldsymbol{\Sigma}_{\xi,s} = \boldsymbol{I}_d$ for all $s$; and the target noise is $\Delta_s = 0.01$ for all $s$. For each subfigure, we modify one data property while keeping the rest same and show its effect on the ICL error: in (a), we focus on the input covariance of the second data source by using $\boldsymbol{\Sigma}_{x,1} = \boldsymbol{I}_d + \theta_x \boldsymbol{\gamma}_x \boldsymbol{\gamma}_x^T$ such that $\|\boldsymbol{\Sigma}_{x,1}\|_2$ is changed by varying $\theta_x$; in (b), we concentrate on the task covariance of the second data source by using $\boldsymbol{\Sigma}_{\xi,1} = \boldsymbol{I}_d + \theta_\xi \boldsymbol{\gamma}_\xi \boldsymbol{\gamma}_\xi^T$ such that $\|\boldsymbol{\Sigma}_{\xi,1}\|_2$ is changed by varying $\theta_\xi$; and in (c), we modify the noise of the second data source $\Delta_1$ while the noise of the first source $\Delta_0$ is set to $0.2$ and fixed. We set the degree of the equivalent polynomial model to $p = 5$. The average of 20 Monte Carlo runs is plotted.

### 4.3.3 Impact of data mixing

We next explore how mixing structured and unstructured data sources affects ICL performance, shown in Figure 2. In these experiments, we vary the second data source by modifying its input covariance (a), task covariance (b), or noise variance (c), while keeping the first source fixed.

 (i) In Figure 2(a), increasing the proportion of samples from a data source with structured inputs significantly improves ICL performance.
 (ii) Figure 2(b) shows a similar trend when the task vectors have structured covariance, with ICL error decreasing as their prevalence increases.
(iii) In Figure 2(c), lowering the noise variance in the second source leads to performance gains, underscoring the sensitivity of ICL to data quality.

When studying the impact of the data mixing, the ICL errors corresponding to each source can be of interest, so we provide them in Appendix G, where we observe behaviors similar to those in Figure 2. Overall, these results demonstrate that Transformers can effectively exploit structured data, and that the relative quality of data sources—defined by structure and noise—directly shapes ICL.

### 4.3.4 Feature learning with data mixing

While prior results in Figures 1- 2 used a fixed step size $\eta \asymp d^2$, we now study the interaction between feature learning and data mixing. Specifically, we assess how varying $\eta$ influences ICL error under two distinct structural conditions: (a) structured input covariance and (b) structured task covariance.

Results in Figure 3(a)-(b) show a notable asymmetry:

 (i) In (a), increasing $\eta$ does not improve ICL error, consistent with our intuition that the first-layer updates do not capture a useful signal when task vectors are isotropic. Here, the intuition comes from the fact that the feature learning enables MLPs to align with task vectors [4, Section 3.2].
 (ii) In contrast, (b) reveals significant performance gains with increasing $\eta$, as the model can extract informative task-related features when the task vectors have structured (non-isotropic) covariance.

This contrast highlights that feature learning via gradient updates primarily enhances representations of task-related directions. The extent of benefit depends on both the structure present in the data and the relative mixing ratios of the sources. These findings emphasize a rich interplay between feature learning dynamics and data heterogeneity in ICL.

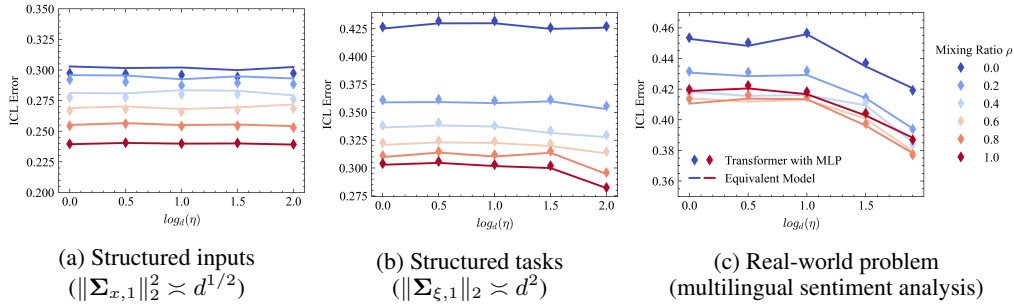

|                                          |                                        |                                                    |
| ---------------------------------------- | -------------------------------------- | -------------------------------------------------- |
| (a) Structured inputs                    | (b) Structured tasks                   | (c) Real-world problem                             |
| $(\|\Sigma_{x,1}\|_2^2 \asymp d^{1/2})$  | $(\|\Sigma_{\xi,1}\|_2 \asymp d^2)$    | (multilingual sentiment analysis)                  |

Figure 3: Feature learning with data mixing on synthetic and real-world data: the effect of different step sizes $\eta$ is illustrated. For the synthetic data scenarios in (a) and (b), we start with the same initial setting as Figure 2. In (a), we just modify the covariance of the input vectors from the second data source, while we only change the covariance of the task vectors from the second data source in (b). In each case, we add an additional rank-one structure to the covariance, which is the same as Figure 2. We plot the average of 20 Monte Carlo trials for (a) and (b). For the real-world data scenario in (c), we focus on the effect of feature learning on ICL errors for multilingual sentiment analysis using the Multilingual Amazon Reviews Corpus [22]. This dataset contains customer reviews (with text and star ratings) in multiple languages, such as English and German. By treating English and German reviews as two distinct data sources, we can vary the mixing ratio across languages, allowing us to evaluate our framework in a more realistic setting. We consider English reviews as source 1 and German reviews as source 0, so a mixing ratio of $\rho = 1$ corresponds to entirely English data, and $\rho = 0$ to entirely German data. For labels $\{y_i\}$, the review star ratings are demeaned and scaled to lie in the range $[-1, 1]$, making the task regression-like. For inputs $\{x_i\}$, each review text is embedded using the multilingual text embedding model called "multilingual-e5-small" [43], the generated 384-dimensional embeddings are reduced to 64 dimensions via PCA (principal component analysis), and then normalized. We group $l$ input-label pairs (of the same language) together to form a context so that we make the problem compatible with our ICL setting. The rest of the details for (c) are $d = l = 64$, $n = k = 0.25d^2$, and $\lambda = 5 \times 10^{-5}$. The degree of the equivalent polynomial model is set to 5. The mean of 100 Monte Carlo trials is illustrated in (c).

### 4.3.5 Data mixing in a real-world scenario: multilingual sentiment analysis

For a real-world example, we consider multilingual sentiment analysis using the Multilingual Amazon Reviews Corpus [22], which contains customer reviews in multiple languages. Treating each language as a distinct source, we examine feature learning under data mixing, as illustrated in Figure 3(c). Consistent with our theoretical predictions, the ICL errors of the Transformer model closely align with those of the equivalent model, even in this multilingual real-world setting. We observe a clear trend (similar to Figure 3(b)) whereby increasing the step size $\eta$ reduces the ICL error, indicating effective feature learning. Finally, a higher proportion of English reviews can lead to increased performance, consistent with the known strength of the embedding model (for text) on English data.

## 5 Conclusion

This work advances the theoretical and empirical understanding of in-context learning (ICL) in pretrained Transformers with nonlinear MLP heads, especially under high-dimensional asymptotics. By leveraging techniques from Gaussian universality and orthogonal polynomials, we establish that such Transformers are effectively equivalent to structured polynomial predictors in terms of ICL error. This equivalence not only clarifies the functional role of nonlinear MLPs in Transformers but also enables precise analysis of their learning behavior. Our findings reveal that nonlinear MLPs substantially enhance ICL on nonlinear tasks, outperforming their linear counterparts. Furthermore, we explored the impact of having a mixture of different data sources during the training, characterizing the properties of a good data source and providing insight about choosing the mixture ratio. Namely, we found that good (high-quality) data sources are those having low target noise and structured covariances for input and task. We showed that feature learning depends on the data mixture, and specifically, having a structured covariance for the task vectors is needed for feature learning. Finally, we illustrated an example showing our findings extend to a real-world scenario involving multilingual sentiment analysis. Overall, this work contributes to the theoretical understanding of the impact of data distribution and feature learning on ICL by Transformers.

## Acknowledgments and Disclosure of Funding

We acknowledge that this work is supported partially by TÜBİTAK under project 124E063 in ARDEB 1001 program. We extend our gratitude to TÜBİTAK for its support. S.D. is supported by an AI Fellowship provided by KUIS AI Center and a PhD Scholarship (BİDEB 2211) from TÜBİTAK. Z.D. is also partially supported by the BİDEB 2224-A program from TÜBİTAK.

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

## Deferred proofs

In the following appendices, we provide deferred proofs. We first consider the case of zero-mean inputs, i.e., $\boldsymbol{\mu}_{x,s} = \mathbf{0}$ for all $s$, and address the extension to non-zero mean inputs, i.e., $\boldsymbol{\mu}_{x,s} \neq \mathbf{0}$, in the final section. This organization streamlines the initial presentation while preserving generality.

Appendix A introduces a reparameterization of the attention mechanism that simplifies the form of the attention outputs. Appendix B analyzes the distribution of $\boldsymbol{F}\mathrm{vec}(\boldsymbol{H_Z})$, a key quantity in our analysis, and proves Lemma 4.9. Appendix C presents a decomposition of the gradient matrix corresponding to a single gradient step on the first layer (6), establishing Lemma 4.11. Appendix D builds on prior results to establish a conditional Gaussian equivalence. Appendix E contains the proof of our main theoretical result (Theorem 4.12). Finally, Appendix F considers an extension of the analysis to the setting with non-zero mean inputs.

## Notation

We adopt standard notation following [18]. For a random vector $\boldsymbol{v}$, its covariance is denoted by $\mathrm{Cov}(\boldsymbol{v})$. The spectral norm and Frobenius norm of a matrix $\boldsymbol{A}$ are denoted by $\|\boldsymbol{A}\|_2$ and $\|\boldsymbol{A}\|_F$, respectively, while $\mathrm{Tr}(\boldsymbol{A})$ denotes the trace. Matrix entries and slices are indicated by $A_{i,j}$, $\boldsymbol{A}_{:,j}$, and $\boldsymbol{A}_{i,:}$, where $i$ and $j$ denote the row and column indices, respectively. $\mathrm{vec}(.)$ denotes the vectorization operation. We write $f(\cdot) \asymp g(\cdot)$ to denote that $f$ and $g$ are of the same asymptotic order with respect to the diverging dimensions. We use standard asymptotic notation: $f(d) = \mathcal{O}(g(d))$ if $f(d)/g(d) \to C < \infty$ for some $C > 0$, and $f(d) = o(g(d))$ if $f(d)/g(d) \to 0$ as $d \to \infty$. Furthermore, we define $\tilde{\mathcal{O}}(f(\cdot))$ as shorthand for $\mathcal{O}(f(\cdot) \,\mathrm{polylog}\, d)$ to suppress polylogarithmic factors for brevity. Element-wise multiplication is denoted by $\odot$, and the Kronecker product by $\otimes$. Conditional random variables are written as $X \mid C$, representing the distribution of $X$ given condition $C$. Finally, we use $x \to \mathcal{N}(0, 1)$ to denote that the random variable $x$ converges in distribution to the standard normal.

## A    Reparameterization of the linear attention

We begin by reparameterizing the linear attention mechanism in accordance with prior work [47, 2, 29, 23, 32, 27] to simplify subsequent analysis. Specifically, we consider the following form of linear attention:

$$\boldsymbol{A} := \boldsymbol{Z} + \frac{1}{\ell}\boldsymbol{V}\boldsymbol{Z}(\boldsymbol{K}\boldsymbol{Z})^T(\boldsymbol{Q}\boldsymbol{Z}), \tag{S1}$$

where $\boldsymbol{K}, \boldsymbol{Q}$, and $\boldsymbol{V}$ denote the key, query, and value matrices, respectively. Due to the structure of the embedding matrix introduced in (2), the output of linear attention, denoted $\hat{y}_{\mathrm{linear}}$, corresponds to the $(d+1, \ell+1)$-th entry of $\boldsymbol{A}$, i.e.,

$$\hat{y}_{\mathrm{linear}} := A_{d+1,\ell+1} = \frac{1}{\ell}\boldsymbol{V}_{d+1,:}(\boldsymbol{Z}\boldsymbol{Z}^T)(\boldsymbol{K}^T\boldsymbol{Q})\boldsymbol{Z}_{:,\ell+1}. \tag{S2}$$

This formulation highlights that the parameters relevant to $\hat{y}_{\mathrm{linear}}$ are $\boldsymbol{V}_{d+1,:}$ and $\boldsymbol{K}^T\boldsymbol{Q}$. To isolate these, we express the matrices as follows:

$$\boldsymbol{V} = \begin{bmatrix} * & * \\ \boldsymbol{v}_{21} & v_{22} \end{bmatrix}, \quad \text{and} \quad \boldsymbol{M} := \boldsymbol{K}^T\boldsymbol{Q} = \begin{bmatrix} \boldsymbol{M}_{11} & * \\ \boldsymbol{m}_{21}^T & * \end{bmatrix}, \tag{S3}$$

where $*$ denotes components not involved in predicting $y_{\ell+1}$, while the submatrices $\boldsymbol{M}_{11} \in \mathbb{R}^{d\times d}$, $\boldsymbol{v}_{21}, \boldsymbol{m}_{21} \in \mathbb{R}^d$, and $v_{22} \in \mathbb{R}$ capture the relevant contributions. Using these components, the prediction $\hat{y}_{\mathrm{linear}}$ can be rewritten in terms of the input features and previous outputs as:

$$\frac{1}{\ell}\left\langle \boldsymbol{x}_{\ell+1}, v_{22}\boldsymbol{M}_{11}^T\sum_{i\leq\ell} y_i\boldsymbol{x}_i + v_{22}\boldsymbol{m}_{21}\sum_{i\leq\ell} y_i^2 + \boldsymbol{M}_{11}^T\sum_{i\leq\ell+1} \boldsymbol{x}_i\boldsymbol{x}_i^T\boldsymbol{v}_{21} + \boldsymbol{m}_{21}\sum_{i\leq\ell} y_i\boldsymbol{x}_i^T\boldsymbol{v}_{21} \right\rangle, \tag{S4}$$

where $\langle \cdot, \cdot \rangle$ is the standard inner product and we omit the source index $s$ for brevity.

In most of the theoretical literature [47, 2, 29, 23, 32], the parameters $\boldsymbol{v}_{21}$ and $\boldsymbol{m}_{21}$ are commonly set to $\mathbf{0}$, since the leading term

$$\frac{1}{\ell} \left\langle \boldsymbol{x}_{\ell+1}, v_{22} \boldsymbol{M}_{11}^T \sum_{i \leq \ell} y_i \boldsymbol{x}_i \right\rangle$$

in (S4) alone is often sufficient to accurately predict $y_{\ell+1}$. More recently, [27] relaxed this assumption by allowing $\boldsymbol{m}_{21}$ to be trainable while keeping $\boldsymbol{v}_{21} = \mathbf{0}$, as this modification retains analytical tractability. Following this approach, we adopt a setting in which $\boldsymbol{v}_{21} = \mathbf{0}$ and the remaining parameters are trainable.

Under this formulation, the output of the linear attention mechanism can be compactly expressed as:

$$\hat{y}_{\text{linear}} = \text{vec}(\boldsymbol{\Gamma})^T \text{vec}(\boldsymbol{H_Z}), \tag{S5}$$

where the trainable parameters are consolidated into the matrix $\boldsymbol{\Gamma}$, and the processed version of the embedding matrix $\boldsymbol{Z}$ is denoted by $\boldsymbol{H_Z}$. These are defined as:

$$\boldsymbol{\Gamma} := v_{22} \begin{bmatrix} \boldsymbol{M}_{11}^T & \boldsymbol{m}_{21} \end{bmatrix}, \quad \text{and} \quad \boldsymbol{H_Z} := \boldsymbol{x}_{\ell+1} \begin{bmatrix} \frac{1}{\ell} \sum_{i \leq \ell} y_i \boldsymbol{x}_i^T & \frac{1}{\ell} \sum_{i \leq \ell} y_i^2 \end{bmatrix}. \tag{S6}$$

The advantage of this reformulation lies in the fact that the linear attention output is now a linear function of both the trainable parameters $\boldsymbol{\Gamma}$ and the data-derived matrix $\boldsymbol{H_Z}$. As a result, the matrix $\boldsymbol{\Gamma}$ can be efficiently optimized via ridge regression.

Similarly, in the case of a Transformer architecture equipped with a nonlinear MLP block, the above formulation enables us to express the prediction as:

$$\hat{y}_{\text{nonlinear}} := \frac{1}{\sqrt{k}} \boldsymbol{w}^T \sigma(\boldsymbol{F} \text{vec}(\boldsymbol{H_Z})), \tag{S7}$$

where $\boldsymbol{F} \in \mathbb{R}^{k \times d(d+1)}$ serves a role analogous to that of $\text{vec}(\boldsymbol{\Gamma})$, with each row of $\boldsymbol{F}$ representing the weights associated with one of the $k$ nonlinear neurons. The function $\sigma$ denotes a nonlinear activation function, allowing the model to capture complex input-output relationships, and $\boldsymbol{w} \in \mathbb{R}^k$ is a trainable vector that linearly combines the outputs of these nonlinear units. Overall, this formulation captures the structure of a Transformer with linear attention followed by a nonlinear (two-layer) MLP block.

## B  Asymptotic distribution of $\boldsymbol{F} \text{vec}(\boldsymbol{H_Z})$

In this section, we analyze the asymptotic distribution of the term $\boldsymbol{F} \text{vec}(\boldsymbol{H_Z})$, which plays a central role in the Transformer with a nonlinear MLP layer. After training the first-layer parameters as described in (6), the prediction of the model (5) takes the form

$$\frac{1}{\sqrt{k}} \boldsymbol{w}^T \sigma\left(\hat{\boldsymbol{F}} \text{vec}(\boldsymbol{H_Z})\right) = \frac{1}{\sqrt{k}} \boldsymbol{w}^T \sigma\left(\boldsymbol{F} \text{vec}(\boldsymbol{H_Z}) + \eta \boldsymbol{G} \text{vec}(\boldsymbol{H_Z})\right),$$

where $\boldsymbol{G}$ denotes the gradient matrix defined in (7). Since $\boldsymbol{F} \text{vec}(\boldsymbol{H_Z})$ is the primary input to the nonlinear activation function $\sigma$, understanding its distribution is a natural and informative starting point.

Let $\boldsymbol{f}_i$ denote the $i$-th row of the matrix $\boldsymbol{F}$, and define $t := \text{Tr}(\text{Cov}(\text{vec}(\boldsymbol{H_Z})))$. For the purpose of this analysis, we initially assume that $\boldsymbol{H_Z}$ is fixed. Under Assumption 4.6, each row $\boldsymbol{f}_i$ is drawn independently from a Gaussian distribution $\mathcal{N}(\mathbf{0}, \boldsymbol{I}_{d(d+1)}/t)$. Consequently, we have

$$\boldsymbol{f}_i^T \text{vec}(\boldsymbol{H_Z}) \sim \mathcal{N}\left(0, \|\text{vec}(\boldsymbol{H_Z})\|_2^2/t\right).$$

Thus, the task of proving Lemma 4.9 reduces to showing that $\|\text{vec}(\boldsymbol{H_Z})\|_2^2/t$ concentrates around 1, which would imply that $\boldsymbol{f}_i^T \text{vec}(\boldsymbol{H_Z}) \to \mathcal{N}(0, 1)$.

To establish this concentration result, we begin with the following lemma, which characterizes $\boldsymbol{H_Z}$ as an outer product of a Gaussian vector and a sub-exponential vector, conditional on the data source index $s$ and the task vector $\boldsymbol{\xi}|s$. The lemma also provides concentration bounds for the norms of these constituent vectors. For brevity, we omit explicit conditioning in some expressions where the context makes it clear.

**Lemma B.1.** *Let $\boldsymbol{b} := \begin{bmatrix} \frac{1}{\ell} \sum_{i \le \ell} y_i \boldsymbol{x}_i \\ \frac{1}{\ell} \sum_{i \le \ell} y_i^2 \end{bmatrix}$, which implies that $\boldsymbol{H_Z} = \boldsymbol{x}_{\ell+1} \boldsymbol{b}^T$. Suppose that the data source index $s$ and the task vector $\boldsymbol{\xi}|s$ are given. Then, conditioned on $s$, the vector $\boldsymbol{x}_{\ell+1}|s$ is Gaussian by definition, and the conditional vector $\boldsymbol{b}|(\boldsymbol{\xi}, s)$ exhibits sub-exponential tails. Furthermore, conditioned on $(s, \boldsymbol{\xi}|s)$, the following concentration bounds hold with high probability:*

$$\left| \|\boldsymbol{x}_{\ell+1}\|_2^2 - \mathrm{Tr}(\mathrm{Cov}(\boldsymbol{x}_{\ell+1})) \right| / \mathrm{Tr}(\mathrm{Cov}(\boldsymbol{x}_{\ell+1})) = o(1), \tag{S8}$$

$$\left| \|\boldsymbol{b}\|_2^2 - \mathrm{Tr}(\mathbb{E}[\boldsymbol{b}\boldsymbol{b}^T]) \right| / \mathrm{Tr}(\mathbb{E}[\boldsymbol{b}\boldsymbol{b}^T]) = o(1). \tag{S9}$$

*Proof.* Given $s$ and $\boldsymbol{\xi}|s$, we have $\boldsymbol{x}_i|s \sim \mathcal{N}(\boldsymbol{\mu}_{x,s}, \boldsymbol{\Sigma}_{x,s})$, and $y_i := \phi_s \left( \frac{(\boldsymbol{\xi}|s)^T (\boldsymbol{x}_i|s)}{\|\boldsymbol{\xi}|s\|_2 \|\boldsymbol{\Sigma}_{x,s}\|_2^{1/2}} \right) + \epsilon_i|s$, where $\phi_s$ is Lipschitz by Assumption 4.7 and $\epsilon_i|s$ is Gaussian by definition.

We can express $\boldsymbol{b}$ as: $\boldsymbol{b} = \frac{1}{\ell} \sum_{i \le \ell} \boldsymbol{b}_i$, where $\boldsymbol{b}_i := \begin{bmatrix} y_i \boldsymbol{x}_i \\ y_i^2 \end{bmatrix}$. Since $\phi_s$ is Lipschitz and $\boldsymbol{x}_i$ is sub-Gaussian, the composition $y_i$ is sub-Gaussian with a constant norm due to Gaussian concentration of Lipschitz functions [40, Theorem 5.2.2].

By Assumption 4.2, we have $\|\boldsymbol{\Sigma}_{x,s}\|_2^2 = \mathcal{O}(d)$ and $\mathrm{Tr}(\boldsymbol{\Sigma}_{x,s}) \asymp d$, so $\boldsymbol{x}_i$ is sub-Gaussian with norm $\mathcal{O}(d^{1/4})$. Then, since the product of sub-Gaussian random variables is sub-exponential [40, Lemma 2.7.7], the term $y_i \boldsymbol{x}_i$ is sub-exponential with norm $\mathcal{O}(d^{1/4})$, and $y_i^2$ is sub-exponential with constant norm. Thus, $\boldsymbol{b}_i$ is a sub-exponential vector with norm $\mathcal{O}(d^{1/4})$.

Applying the (vector version of) Bernstein's inequality [40, Corollary 2.8.3] to the vector $\boldsymbol{b}$ (an average over $l$ samples of a sub-exponential) implies that the deviation $(\boldsymbol{b} - \mathbb{E}[\boldsymbol{b}])$ has sub-exponential tails with norm $\mathcal{O}(d^{1/4}/\ell)$. Under Assumption 4.1, $\ell/d \in \mathbb{R}$, and hence the norm becomes $\mathcal{O}(d^{-3/4})$.

Finally, applying the Hanson-Wright inequality [40, Theorem 6.2.1] yields the bound in (S8). For (S9), we rely on an extension of the Hanson-Wright inequality to sub-exponential vectors [19]. $\square$

Now that we have bounds on the norms of the vectors used to construct $\boldsymbol{H_Z}$, we can analyze the concentration of the norm of $\mathrm{vec}(\boldsymbol{H_Z})$ in the following corollary, which builds on Lemma B.1.

**Corollary B.2.** *With high probability, the following concentration result holds:*

$$\frac{\left| \|vec(\boldsymbol{H_Z}) \,|\, (s, \boldsymbol{\xi}|s)\|_2^2 - \mathrm{Tr}(\mathrm{Cov}(vec(\boldsymbol{H_Z}) \,|\, (s, \boldsymbol{\xi}|s))) \right|}{\mathrm{Tr}(\mathrm{Cov}(vec(\boldsymbol{H_Z}) \,|\, (s, \boldsymbol{\xi}|s)))} = \left| \frac{\|vec(\boldsymbol{H_Z}) \,|\, (s, \boldsymbol{\xi}|s)\|_2^2}{t} - 1 \right| = o(1),$$

*where $t := \mathrm{Tr}(\mathrm{Cov}(vec(\boldsymbol{H_Z})))$.*

*Proof.* We first note that:

$$\mathrm{vec}(\boldsymbol{H_Z}) = \mathrm{vec}(\boldsymbol{x}_{\ell+1} \boldsymbol{b}^T) = \boldsymbol{b} \otimes \boldsymbol{x}_{\ell+1}.$$

Using this, we compute the covariance:

$$\mathrm{Cov}(\mathrm{vec}(\boldsymbol{H_Z})) = \mathbb{E}[(\boldsymbol{b} \otimes \boldsymbol{x}_{\ell+1})(\boldsymbol{b} \otimes \boldsymbol{x}_{\ell+1})^T], \tag{S10}$$

$$= \mathbb{E}[(\boldsymbol{b} \otimes \boldsymbol{x}_{\ell+1})(\boldsymbol{b}^T \otimes \boldsymbol{x}_{\ell+1}^T)], \tag{S11}$$

$$= \mathbb{E}[(\boldsymbol{b}\boldsymbol{b}^T) \otimes (\boldsymbol{x}_{\ell+1} \boldsymbol{x}_{\ell+1}^T)], \tag{S12}$$

$$= \mathbb{E}[\boldsymbol{b}\boldsymbol{b}^T] \otimes \mathrm{Cov}(\boldsymbol{x}_{\ell+1}), \tag{S13}$$

where we used properties of the Kronecker product (transpose and mixed-product), the independence of $\boldsymbol{b}$ and $\boldsymbol{x}_{\ell+1}$, and the linearity of expectation.

Similarly, we have:

$$\|\mathrm{vec}(\boldsymbol{H_Z})\|_2^2 = \|\boldsymbol{b} \otimes \boldsymbol{x}_{\ell+1}\|_2^2 = \|\boldsymbol{b}\|_2^2 \cdot \|\boldsymbol{x}_{\ell+1}\|_2^2, \tag{S14}$$

and

$$\mathrm{Tr}(\mathrm{Cov}(\mathrm{vec}(\boldsymbol{H_Z}))) = \mathrm{Tr}(\mathbb{E}[\boldsymbol{b}\boldsymbol{b}^T]) \cdot \mathrm{Tr}(\mathrm{Cov}(\boldsymbol{x}_{\ell+1})). \tag{S15}$$

Define the relative errors:

$$\delta_x := \frac{\left|\|\boldsymbol{x}_{\ell+1}\|_2^2 - \text{Tr}(\text{Cov}(\boldsymbol{x}_{\ell+1}))\right|}{\text{Tr}(\text{Cov}(\boldsymbol{x}_{\ell+1}))}, \quad \text{and} \quad \delta_v := \frac{\left|\|\boldsymbol{b}\|_2^2 - \text{Tr}(\mathbb{E}[\boldsymbol{b}\boldsymbol{b}^T])\right|}{\text{Tr}(\mathbb{E}[\boldsymbol{b}\boldsymbol{b}^T])}. \tag{S16}$$

Then, conditioned on $(s, \boldsymbol{\xi}|s)$, we get:

$$\frac{\left|\|\text{vec}(\boldsymbol{H_Z})\|_2^2 - \text{Tr}(\text{Cov}(\text{vec}(\boldsymbol{H_Z})))\right|}{\text{Tr}(\text{Cov}(\text{vec}(\boldsymbol{H_Z})))} = \frac{\left|\|\boldsymbol{x}_{\ell+1}\|_2^2\|\boldsymbol{b}\|_2^2 - \text{Tr}(\text{Cov}(\boldsymbol{x}_{\ell+1}))\text{Tr}(\mathbb{E}[\boldsymbol{b}\boldsymbol{b}^T])\right|}{\text{Tr}(\text{Cov}(\boldsymbol{x}_{\ell+1}))\text{Tr}(\mathbb{E}[\boldsymbol{b}\boldsymbol{b}^T])} \tag{S17}$$

$$\leq \delta_x + \delta_v + \delta_x\delta_v \tag{S18}$$

$$= o(1), \tag{S19}$$

where the final step follows from Lemma B.1, which showed that both $\delta_x$ and $\delta_v$ are $o(1)$. $\qquad\square$

So far, our analysis has focused on the case where $\boldsymbol{H_Z}$ is conditioned on the data source $s$ and the task vector $\boldsymbol{\xi}|s$. However, by Assumption 4.4, we have

$$t = \text{Tr}(\text{Cov}(\text{vec}(\boldsymbol{H_Z}) \mid s = i)) = \text{Tr}(\text{Cov}(\text{vec}(\boldsymbol{H_Z}) \mid s = j))$$

for any data source indices $i, j$. Furthermore, Corollary B.2 showed that $\|\text{vec}(\boldsymbol{H_Z})\|_2^2/t$ concentrates around 1 for all $s$ and $\boldsymbol{\xi}|s$. This leads us to the following unconditioned concentration result.

**Corollary B.3.** *Without conditioning on the data source or task vector, the following holds with high probability:*

$$\left|\frac{\|vec(\boldsymbol{H_Z})\|_2^2}{t} - 1\right| \leq \max_{s,\boldsymbol{\xi}|s}\left|\frac{\|vec(\boldsymbol{H_Z}) \mid (s,\boldsymbol{\xi}|s)\|_2^2}{t} - 1\right| = o(1), \tag{S20}$$

*as a consequence of Lemma B.1 and Corollary B.2.*

*Proof.* We apply a union bound over the (finite) set of data sources and integrate over the distribution of $\boldsymbol{\xi}|s$. Since the conditional deviation is uniformly bounded by $o(1)$, the same bound extends to the unconditioned case. $\qquad\square$

*Remark* B.4. (Implications of task vector randomness) Corollary B.3 implies that the variability introduced by the task vector $\boldsymbol{\xi}|s$ does not significantly affect the norm of the attention output $\text{vec}(\boldsymbol{H_Z})$, with deviations bounded by $o(1)$. This concentration result justifies a simplification in our analysis: we may treat $\boldsymbol{\xi}|s$ as effectively fixed and analyze the system under a worst-case $\boldsymbol{\xi}|s$, rather than integrating over its full distribution. This reduces analytical complexity while preserving rigorous control over variation across tasks. In effect, the randomness of task vectors can be absorbed into a uniform bound, allowing our results to hold uniformly over task diversity.

This completes our analysis of the concentration of $\|\text{vec}(\boldsymbol{H_Z})\|_2^2/t$ around 1, and thereby concludes the proof of Lemma 4.9.

## C Decomposition of the gradient matrix

Having characterized the distribution of $\boldsymbol{F}\text{vec}(\boldsymbol{H_Z})$, we now turn our attention to the effect of a single gradient step during training, as defined in Equation (6). Specifically, we analyze the structure of the gradient matrix $\boldsymbol{G}$, following the approach of [4, 13].

Recall the definition of the gradient matrix (7):

$$\boldsymbol{G} := \frac{1}{n}\left(\frac{1}{\sqrt{k}}\left(\boldsymbol{w}\tilde{\boldsymbol{y}}^T - \frac{1}{\sqrt{k}}\boldsymbol{w}\boldsymbol{w}^T\sigma(\boldsymbol{F}\tilde{\boldsymbol{H}}^T)\right) \odot \sigma'(\boldsymbol{F}\tilde{\boldsymbol{H}}^T)\right)\tilde{\boldsymbol{H}}. \tag{S21}$$

A complication arises from the presence of the derivative of the activation function $\sigma$, which appears as a multiplicative factor. To simplify this expression, we apply an orthogonal decomposition to the derivative:

$$\sigma'(z) = \alpha + \sigma'_\perp(z),$$

where $\alpha := \mathbb{E}_{z \sim \mathcal{N}(0,1)}[\sigma'(z)]$ is the average slope, and $\sigma'_\perp(z) := \sigma'(z) - \alpha$ captures the zero-mean deviation. The decomposition is justified by the fact that the random variable entering $\sigma$—namely, $\boldsymbol{f}_i^T \tilde{\boldsymbol{h}}_j$—converges in distribution to $\mathcal{N}(0,1)$ for all $i, j$, by Lemma 4.9.

Using this decomposition, the gradient matrix $\boldsymbol{G}$ can be rewritten as:

$$\boldsymbol{G} = \underbrace{\boldsymbol{u}\boldsymbol{v}^T}_{\text{spike}} + \underbrace{\frac{1}{n\sqrt{k}}\left(\boldsymbol{w}\tilde{\boldsymbol{y}}^T \odot \sigma'_\perp(\boldsymbol{F}\tilde{\boldsymbol{H}}^T)\right)\tilde{\boldsymbol{H}} - \frac{1}{nk}\left(\boldsymbol{w}\boldsymbol{w}^T \sigma(\boldsymbol{F}\tilde{\boldsymbol{H}}^T) \odot \sigma'(\boldsymbol{F}\tilde{\boldsymbol{H}}^T)\right)\tilde{\boldsymbol{H}}}_{\Delta}, \quad \text{(S22)}$$

where we define $\boldsymbol{u} := \alpha\boldsymbol{w}$ and $\boldsymbol{v} := \tilde{\boldsymbol{H}}^T\tilde{\boldsymbol{y}}/(n\sqrt{k})$.

This decomposition separates the gradient into two components: A *spike term* $\boldsymbol{u}\boldsymbol{v}^T$, which is expected to dominate, and a *residual term* $\Delta$.

Our next goal is to establish that $\|\boldsymbol{u}\boldsymbol{v}^T\|_2 \gg \|\Delta\|_2$, thereby confirming that the spike term governs the spectral structure of $\boldsymbol{G}$.

A new technical challenge arises in this context: unlike prior work [4, 13], where $\tilde{\boldsymbol{H}}$ is composed of i.i.d. Gaussian vectors, in our setting each row of $\tilde{\boldsymbol{H}}$ is a realization of $\text{vec}(\boldsymbol{H_Z})$. As shown in Lemma B.1 (Appendix B), $\boldsymbol{H_Z}$ is the outer product of a Gaussian vector and a sub-exponential vector, making $\text{vec}(\boldsymbol{H_Z})$ a heavy-tailed [41] random vector. This deviates from the sub-Gaussian assumptions in prior analyses and introduces additional complexity in bounding the norms of $\tilde{\boldsymbol{H}}$.

Nevertheless, in the following lemma, we characterize a spectral norm bound for $\tilde{\boldsymbol{H}}$ that accommodates this heavy-tailed structure.

**Lemma C.1.** *Under our assumptions, the spectral norm of the matrix $\tilde{\boldsymbol{H}}$ satisfies the following bound with high probability:*

$$\|\tilde{\boldsymbol{H}}\|_2 \,/\, \|\text{Cov}(vec(\boldsymbol{H_Z}))\|_2^{1/2} = \tilde{\mathcal{O}}(d). \tag{S23}$$

*Proof.* Corollaries B.2 and B.3 imply that

$$\left|\|\text{vec}(\boldsymbol{H_Z})\|_2^2 - \text{Tr}(\text{Cov}(\text{vec}(\boldsymbol{H_Z})))\right| = \mathcal{O}(d^2)$$

with high probability, given that $\text{Tr}(\text{Cov}(\text{vec}(\boldsymbol{H_Z}))) \asymp d^2$ by Assumption 4.4. It follows that

$$\|\text{vec}(\boldsymbol{H_Z})\|_2 = \mathcal{O}(d)$$

holds with high probability. We now apply a result from [39, Theorem 5.44] concerning the spectral norm of a random matrix with independent heavy-tailed rows. This yields the stated bound on $\|\tilde{\boldsymbol{H}}\|_2$. Note that the assumption $n/d^2 \in \mathbb{R}$ (from Assumption 4.1) ensures the validity of this application. $\square$

We now establish high-probability norm bounds for other key random quantities involved in the analysis. Specifically, the following lemma bounds the norms of $\boldsymbol{w}$, $\boldsymbol{F}$, and $\tilde{\boldsymbol{y}}$.

**Lemma C.2.** *Under our assumptions, the following bounds hold with high probability:*

*(i)* $\|\boldsymbol{w}\|_2 = \tilde{\mathcal{O}}(1)$ *and* $\|\boldsymbol{w}\|_\infty = \tilde{\mathcal{O}}(k^{-1/2})$,

*(ii)* $\|\boldsymbol{F}\|_2 = \tilde{\mathcal{O}}(1)$,

*(iii)* $\|\tilde{\boldsymbol{y}}\|_2 = \tilde{\mathcal{O}}(n^{1/2})$ *and* $\|\tilde{\boldsymbol{y}}\|_\infty = \tilde{\mathcal{O}}(1)$,

*where $k, n = \mathcal{O}(d^2)$ as specified by Assumption 4.1.*

*Proof.* **(i)** Follows from standard sub-Gaussian norm bounds, specifically [40, Proposition 2.5.2 and Theorem 3.1.1], in conjunction with Assumption 4.6.

**(ii)** The spectral norm bound on $\boldsymbol{F}$ under Assumption 4.6 is a direct consequence of the concentration of the spectral norm for sub-Gaussian random matrices; see [40, Theorem 4.4.5].

**(iii)** The bounds for $\tilde{\boldsymbol{y}}$ follow from the Gaussian concentration of Lipschitz functions [40, Theorem 5.2.2], noting that each element of $\tilde{\boldsymbol{y}}$ is defined via a Lipschitz transformation of sub-Gaussian variables. $\square$

Next, we derive high-probability norm bounds for key quantities appearing in the residual term $\boldsymbol{\Delta}$ of the decomposed gradient expression in (S22). Specifically, we consider the terms $\sigma'(\boldsymbol{F}\tilde{\boldsymbol{H}}^T)$, $\sigma'_\perp(\boldsymbol{F}\tilde{\boldsymbol{H}}^T)$, and $\boldsymbol{w}^T\sigma(\boldsymbol{F}\tilde{\boldsymbol{H}}^T)$.

**Lemma C.3.** *Under our assumptions, the following bounds hold with high probability:*

   *(i)* $\|\sigma'_\perp(\boldsymbol{F}\tilde{\boldsymbol{H}}^T)\|_2 \, / \, \|\tilde{\boldsymbol{H}}\|_2 = \tilde{\mathcal{O}}(1)$,

   *(ii)* $\|\sigma'(\boldsymbol{F}\tilde{\boldsymbol{H}}^T)\|_2 = \tilde{\mathcal{O}}(k)$,

   *(iii)* $\|\boldsymbol{w}^T\sigma(\boldsymbol{F}\tilde{\boldsymbol{H}}^T)\|_\infty \, / \, \|\tilde{\boldsymbol{H}}\|_2 = \tilde{\mathcal{O}}(k^{-1/2})$,

*where $k = \mathcal{O}(d^2)$, as specified by Assumption 4.1.*

*Proof.* **(i)** Condition on $\tilde{\boldsymbol{H}}$. By construction, the rows of $\boldsymbol{F}\tilde{\boldsymbol{H}}^T$ are independent centered Gaussian vectors (with generally anisotropic covariance inherited from $\mathrm{Cov}(\mathrm{vec}(\boldsymbol{H_Z}))$). The function $\sigma'_\perp(\cdot)$ is bounded and Lipschitz by Assumption 4.8 so we apply Gaussian concentration for Lipschitz functions [40, Theorem 5.2.2]. This gives us the rows of $\sigma'_\perp(\boldsymbol{F}\tilde{\boldsymbol{H}}^T)$ as independent sub-Gaussian vectors with sub-Gaussian norm proportional to $\|\tilde{\boldsymbol{H}}\|_2 \mathrm{Tr}(\mathrm{Cov}(\mathrm{vec}(\boldsymbol{H_Z})))^{-1/2}$ with $\mathrm{Tr}(\mathrm{Cov}(\mathrm{vec}(\boldsymbol{H_Z}))) \asymp k$ by Assumption 4.4. Finally, employing the spectral norm bound for matrices with independent sub-Gaussian rows [39, Theorem 5.39 and Eq. (5.26)], and then removing the conditioning, yields the stated bound.

**(ii)** We bound the full derivative term as

$$\|\sigma'(\boldsymbol{F}\tilde{\boldsymbol{H}}^T)\|_2 \leq \|\sigma'_\perp(\boldsymbol{F}\tilde{\boldsymbol{H}}^T)\|_2 + \alpha\|\mathbf{1}_{k\times n}\|_2,$$

where $\alpha := \mathbb{E}_{z\sim\mathcal{N}(0,1)}[\sigma'(z)]$ and $\mathbf{1}_{k\times n}$ denotes a matrix of all ones. Since $\|\mathbf{1}_{k\times n}\|_2 = \sqrt{kn} = \tilde{\mathcal{O}}(k)$ under Assumption 4.1, the result follows from part (i).

**(iii)** Similar to (i), using Gaussian concentration of Lipschitz functions [40, Theorem 5.2.2] along with $\|\boldsymbol{w}\|_2 = \tilde{\mathcal{O}}(1)$ from Lemma C.2 gives the claim. $\qquad\square$

With all necessary high-probability bounds in place, we now derive a bound on the norm of the residual term $\boldsymbol{\Delta}$ appearing in the gradient decomposition. Recall:

$$\|\boldsymbol{\Delta}\|_2 = \left\|\frac{1}{n\sqrt{k}}\left(\boldsymbol{w}\tilde{\boldsymbol{y}}^T \odot \sigma'_\perp(\boldsymbol{F}\tilde{\boldsymbol{H}}^T)\right)\tilde{\boldsymbol{H}} - \frac{1}{nk}\left(\boldsymbol{w}\boldsymbol{w}^T\sigma(\boldsymbol{F}\tilde{\boldsymbol{H}}^T) \odot \sigma'(\boldsymbol{F}\tilde{\boldsymbol{H}}^T)\right)\tilde{\boldsymbol{H}}\right\|_2, \quad\text{(S24)}$$

$$\leq \frac{1}{n\sqrt{k}}\left\|\boldsymbol{w}\tilde{\boldsymbol{y}}^T \odot \sigma'_\perp(\boldsymbol{F}\tilde{\boldsymbol{H}}^T)\right\|_2 \|\tilde{\boldsymbol{H}}\|_2 + \frac{1}{nk}\left\|\boldsymbol{w}\boldsymbol{w}^T\sigma(\boldsymbol{F}\tilde{\boldsymbol{H}}^T) \odot \sigma'(\boldsymbol{F}\tilde{\boldsymbol{H}}^T)\right\|_2 \|\tilde{\boldsymbol{H}}\|_2, \quad\text{(S25)}$$

$$\leq \frac{1}{n\sqrt{k}}\|\boldsymbol{w}\|_\infty\|\tilde{\boldsymbol{y}}\|_\infty\|\sigma'_\perp(\boldsymbol{F}\tilde{\boldsymbol{H}}^T)\|_2\|\tilde{\boldsymbol{H}}\|_2 + \frac{1}{nk}\|\boldsymbol{w}\|_\infty\|\boldsymbol{w}^T\sigma(\boldsymbol{F}\tilde{\boldsymbol{H}}^T)\|_\infty\|\sigma'(\boldsymbol{F}\tilde{\boldsymbol{H}}^T)\|_2\|\tilde{\boldsymbol{H}}\|_2, \quad\text{(S26)}$$

$$= \tilde{\mathcal{O}}(k^{-\beta}), \quad\text{(S27)}$$

which holds with high probability, where $\beta \in (0, 1)$ satisfies $\|\mathrm{Cov}(\mathrm{vec}(\boldsymbol{H_Z}))\|_2 = \mathcal{O}(k^{-\beta+1})$ and $\eta = o(k^\beta)$ under Assumption 4.5.

To derive this bound, we first apply the triangle inequality, and then use the identity for Hadamard products:

$$\boldsymbol{a}\boldsymbol{b}^T \odot \boldsymbol{C} = \mathrm{diag}(\boldsymbol{a})\,\boldsymbol{C}\,\mathrm{diag}(\boldsymbol{b}),$$

which allows us to factor out the vectors and simplify norm computations. The final result follows by applying high-probability bounds established in Lemmas C.1, C.2, and C.3.

Similarly, we can bound the norm of the spiked term $\boldsymbol{u}\boldsymbol{v}^T$:

$$\|\boldsymbol{u}\boldsymbol{v}^T\|_2 = \|\boldsymbol{u}\|_2\|\boldsymbol{v}\|_2 = \frac{\alpha}{n\sqrt{k}}\|\boldsymbol{w}\|_2\|\tilde{\boldsymbol{H}}\|_2\|\tilde{\boldsymbol{y}}\|_2 = \tilde{\mathcal{O}}(k^{-\beta/2}), \quad\text{(S28)}$$

which also holds with high probability.

Taken together, the bounds in (S27) and (S28) confirm that the leading contribution to the gradient matrix arises from the rank-one term $uv^T$, whereas $\Delta$ constitutes a negligible residual. This completes the proof of Lemma 4.11.

# D   Conditional Gaussian equivalence

With the results from the preceding appendices, we now establish a new result: a *conditional Gaussian equivalence*, which will play a key role in the proof of Theorem 4.12. To formulate this result, we first define a subspace relevant for the conditioning in the equivalence argument.

**Lemma D.1** (Decomposition of $\hat{F}\mathrm{vec}(H_Z)$ conditioned on data source $s$). *Let $s$ be a fixed data source. Define the subspace*
$$S := [v, \gamma_{s,1}, \gamma_{s,2}, \ldots, \gamma_{s,r_s}],$$
*where $v$ is the spiked direction from Lemma 4.11, and $\gamma_{s,1}, \ldots, \gamma_{s,r_s}$ are the spiked directions of $\mathrm{Cov}(\mathrm{vec}(H_Z))$ as specified in Assumption 4.4. Let $P$ and $P_\perp$ denote the orthogonal projection matrices onto $\mathrm{span}(S)$ and its orthogonal complement, respectively. Then,*
$$\hat{F}vec(H_Z) = (F + \eta\Delta)P_\perp vec(H_Z) + \hat{F}S\kappa_s, \tag{S29}$$
*where $\kappa_s := (S^T S)^{-1} S^T vec(H_Z)$.*

*Proof.* This decomposition follows directly from the definition of orthogonal projections and the linearity of matrix multiplication. ∎

Lemma D.1 yields a decomposition of the transformed hidden state $\hat{F}\mathrm{vec}(H_Z)$ into two components: a "bulk" term that is conditionally Gaussian, and a structured term aligned with the low-rank subspace $S$ induced by spiked directions. Crucially, this implies that, conditional on the coefficient vector $\kappa_s$, the transformation $\hat{F}\mathrm{vec}(H_Z)$ is sub-Gaussian. This motivates the following conditional Gaussian equivalence theorem, which approximates the nonlinear feature map $\sigma(\hat{F}\mathrm{vec}(H_Z))$ with a Gaussian counterpart conditioned on $(s, \kappa_s)$.

**Theorem D.2** (Conditional Gaussian equivalence). *Under the assumptions in Section 4.1, define the nonlinear feature map $\psi(vec(H_Z)) := \sigma(\hat{F}vec(H_Z))$, and let $o := P_\perp vec(H_Z)$ be the projection onto the orthogonal complement of the subspace $S$ defined in Lemma D.1. Then, the following conditional Gaussian feature map is equivalent to $\psi(vec(H_Z))$ in terms of ICL error, when conditioned on data source $s$ and alignment vector $\kappa_s$:*
$$\hat{\psi}(vec(H_Z); s, \kappa_s) := \nu(s, \kappa_s) + \Psi(s, \kappa_s)o + \Phi(s, \kappa_s)^{1/2}g, \tag{S30}$$
*where $g \sim \mathcal{N}(0, I)$, and*
$$\nu(s, \kappa_s) := \mathbb{E}\left[\sigma(\hat{F}vec(H_Z)) \,\Big|\, s, \kappa_s\right],$$
$$\Psi(s, \kappa_s) := \mathbb{E}\left[\sigma(\hat{F}vec(H_Z))o^T \,\Big|\, s, \kappa_s\right],$$
$$\Phi(s, \kappa_s) := \mathrm{Cov}\left(\sigma(\hat{F}vec(H_Z)) \,\Big|\, s, \kappa_s\right) - \Psi(s, \kappa_s)\Psi(s, \kappa_s)^T.$$

*Proof.* The proof strategy follows the standard approach to Gaussian equivalence for random features as developed in [31, 11, 10, 13]. In particular, it is sufficient to establish a central limit theorem (CLT) for the bulk component $(F + \eta\Delta)o$, conditional on $s, \kappa_s$. Once this CLT is in place, the remainder of the proof mirrors that of [13], and is omitted here for brevity.

**Conditional CLT**   For any Lipschitz function $\zeta : \mathbb{R}^2 \to \mathbb{R}$, for all $s \in \{0, \ldots, \mathcal{S} - 1\}$ and for all $\kappa_s \in \mathbb{R}^{r_s+1}$,

$$\lim_{d,k\to\infty} \sup_{\tilde{w},\tilde{\xi}} \left| \mathbb{E}\left[ \zeta\left(\tilde{w}^T\psi(\mathrm{vec}(H_Z)), \tilde{\xi}^T x\right) \,\Big|\, s, \kappa_s\right] - \mathbb{E}\left[ \zeta\left(\tilde{w}^T\hat{\psi}(\mathrm{vec}(H_Z)), \tilde{\xi}^T x\right) \,\Big|\, s, \kappa_s\right] \right| = 0,$$
$$\tag{S31}$$

where the supremum is taken over $\tilde{\boldsymbol{w}} \in \{\boldsymbol{w} \in \mathbb{R}^k \mid \|\boldsymbol{w}\|_2 = \mathcal{O}(1), \|\boldsymbol{w}\|_\infty = \mathcal{O}(k^{-\epsilon})\}$ for some $\epsilon > 0$, and $\tilde{\boldsymbol{\xi}} \in \mathbb{R}^d$ satisfies $\|\tilde{\boldsymbol{\xi}}\|_2 = 1/\|\mathrm{Cov}(\boldsymbol{x} \mid s)\|_2^{1/2}$. Here, $k, d \to \infty$ such that $k/d^2 \in \mathbb{R}$, as specified in Assumption 4.1.

This conditional CLT establishes the equivalence of the original and Gaussian feature maps, $\psi(\mathrm{vec}(\boldsymbol{H_Z}))$ and $\hat{\psi}(\mathrm{vec}(\boldsymbol{H_Z}); s, \boldsymbol{\kappa}_s)$, in terms of their behavior under any test function $\zeta$, conditional on $s$ and $\boldsymbol{\kappa}_s$. Notably, the supremum ensures robustness to variation in task vectors, accounting for worst-case alignment.

We begin proving the conditional CLT by recalling the decomposition from Lemma D.1:

$$\hat{\boldsymbol{F}}\mathrm{vec}(\boldsymbol{H_Z}) = (\boldsymbol{F} + \eta\boldsymbol{\Delta})\boldsymbol{P}_\perp\mathrm{vec}(\boldsymbol{H_Z}) + \hat{\boldsymbol{F}}\boldsymbol{S}\boldsymbol{\kappa}_s.$$

Define $\boldsymbol{r} := (\boldsymbol{F} + \eta\boldsymbol{\Delta})\boldsymbol{P}_\perp\mathrm{vec}(\boldsymbol{H_Z})$ and $\boldsymbol{c} := \hat{\boldsymbol{F}}\boldsymbol{S}\boldsymbol{\kappa}_s$ so that $\hat{\boldsymbol{F}}\mathrm{vec}(\boldsymbol{H_Z}) = \boldsymbol{r} + \boldsymbol{c}$. The random vector $\boldsymbol{r}$ is approximately Gaussian because:

- $\eta\|\boldsymbol{\Delta}\|_2 \to 0$ asymptotically (see Appendix C),

- $\boldsymbol{P}_\perp$ is an orthogonal projection ($\|\boldsymbol{P}_\perp\|_2 = 1$),

- Lemma 4.9 shows that $\boldsymbol{F}\mathrm{vec}(\boldsymbol{H_Z})$ converges to a Gaussian distribution,

- Assumption 4.4 ensures $\boldsymbol{P}_\perp$ eliminates spiked directions, leading to $\boldsymbol{r} \to \mathcal{N}(0, \boldsymbol{I}_k)$.

Consequently, we may write $\boldsymbol{r} \stackrel{d}{=} \tilde{\boldsymbol{F}}\boldsymbol{q}$, where $\stackrel{d}{=}$ denotes equivalence in distribution, $\tilde{\boldsymbol{F}}$ is a random feature matrix satisfying the conditions in [21] and $\boldsymbol{q} \sim \mathcal{N}(0, \boldsymbol{I})$. Meanwhile, $\boldsymbol{c}$ is deterministic conditional on $(\boldsymbol{F}, s, \boldsymbol{\kappa}_s)$. Define the neuron-wise conditional activation:

$$\tilde{\sigma}_{j|(s,\boldsymbol{\kappa}_s)}(\tilde{\boldsymbol{f}}_j^T\boldsymbol{q}) := \sigma_j(\tilde{\boldsymbol{f}}_j^T\boldsymbol{q} + c_j), \quad j = 1, \dots, k.$$

This representation allows us to view the feature map as a collection of neuron-specific activations with fixed shifts $c_j$. Importantly, the CLT for random features in [21] applies even when activations vary across neurons, a point further supported by related results in [11, 13].

Utilizing Corollary 4.10 and applying the central limit theorem for heterogeneous neuron activations [21, Theorem 2], we obtain convergence in distribution of $\psi(\mathrm{vec}(\boldsymbol{H_Z}))$ to its Gaussian approximation $\hat{\psi}(\mathrm{vec}(\boldsymbol{H_Z}); s, \boldsymbol{\kappa}_s)$. Furthermore, the odd activation function assumption required in [21] can be omitted here, since both feature maps share the same conditional covariance structure. See [31, 11, 10, 13] for related proofs and further technical details.

So far, we have established an equivalence with respect to the training error. While not stated explicitly earlier, this training error corresponds to that of ridge regression applied to the second-layer weights in our two-stage training setup. To extend this asymptotic equivalence to the ICL error defined in (8)—which is our main object of interest—we require an additional technical condition, formalized below.

**Assumption D.3.** Consider a perturbed optimization objective:

$$\mathcal{T}_n(c) := \min_{\boldsymbol{w} \in \mathcal{C}_k} \frac{1}{n} \sum_{j=1}^n \left( y_{\ell+1}^j - \frac{1}{\sqrt{k}}\boldsymbol{w}^T\sigma(\hat{\boldsymbol{F}}\mathrm{vec}(\boldsymbol{H}_{\boldsymbol{Z}^j})) \right)^2 + \lambda\|\boldsymbol{w}\|_2^2 + c\,\mathcal{E}(\boldsymbol{w}), \tag{S32}$$

where $c \in \mathbb{R}$ is a scalar perturbation parameter, and $\mathcal{E}(\boldsymbol{w})$ denotes the ICL error from (8) associated with second-layer weight vector $\boldsymbol{w}$. The constraint set $\mathcal{C}_k$ is the constraint set in the conditional CLT and it is defined as $\mathcal{C}_k := \{\boldsymbol{w} \in \mathbb{R}^k \mid \|\boldsymbol{w}\| = \mathcal{O}(1), \|\boldsymbol{w}\|_\infty = \mathcal{O}(k^{-\epsilon})\}$ for some $\epsilon > 0$. Then, there exists a constant $c^* > 0$ such that, for all $c \in [-c^*, c^*]$, the function $\mathcal{T}_n(c)$ converges pointwise to a limiting function $\mathcal{T}(c)$, which is differentiable at $c = 0$.

Although this assumption may appear somewhat artificial at first glance, it enables the use of convexity-based arguments in establishing generalization error equivalence (ICL errors in our case), as seen in prior work on Gaussian universality [11, Assumption 5]. Similar assumptions are also employed in related studies on asymptotic equivalence [21, 31, 13]. Importantly, this condition arises primarily as a technical requirement of the proof method, rather than as a limitation on the broader applicability of the Gaussian equivalence results established in this work.

With Assumption D.3 in place, we are now able to extend the asymptotic equivalence result to the ICL error, thereby completing the proof of the conditional Gaussian equivalence, as established in [13]. □

Theorem D.2 establishes an asymptotic equivalence between two feature maps with respect to the ICL error, under the condition that their first two conditional moments match. This result implies the existence of an equivalent (conditional) Gaussian model—namely, $\boldsymbol{w}^T \hat{\psi}(\text{vec}(\boldsymbol{H_Z}); s, \boldsymbol{\kappa}_s)$—that can replace the original feature map without affecting ICL performance. Leveraging this equivalence, we now prove that the Transformer model with a nonlinear MLP is asymptotically equivalent to a polynomial model, as formalized in Theorem 4.12. The full proof is presented below.

# E  Equivalent polynomial model

We aim to establish the asymptotic equivalence between a Transformer model with a nonlinear MLP and the polynomial model described in Theorem 4.12. Our strategy relies on the conditional Gaussian equivalence result stated and proved in Appendix D (Theorem D.2). This result asserts that two models with matching conditional means and covariances yield the same ICL error (8). Thus, to prove equivalence, it suffices to demonstrate that the first two conditional moments of the two models coincide.

Fix a data source index $s$, and recall the orthogonal decomposition with respect to the subspace defined by the matrix $\boldsymbol{S}$:

$$\hat{\boldsymbol{F}}\text{vec}(\boldsymbol{H_Z}) = (\boldsymbol{F} + \eta\boldsymbol{\Delta})\boldsymbol{P}_\perp \text{vec}(\boldsymbol{H_Z}) + \hat{\boldsymbol{F}}\boldsymbol{S}\boldsymbol{\kappa}_s, \tag{S33}$$

where $\boldsymbol{\kappa}_s := (\boldsymbol{S}^T\boldsymbol{S})^{-1}\boldsymbol{S}^T\text{vec}(\boldsymbol{H_Z})$, as established in Lemma D.1. The first term, $(\boldsymbol{F} + \eta\boldsymbol{\Delta})\boldsymbol{P}_\perp \text{vec}(\boldsymbol{H_Z})$, behaves asymptotically like a Gaussian random variable (see the proof of Theorem D.2), while the second term, $\hat{\boldsymbol{F}}\boldsymbol{S}\boldsymbol{\kappa}_s$, is deterministic conditional on $\boldsymbol{F}$, $s$, and $\boldsymbol{\kappa}_s$.

According to [13, Theorem 4], if conditional Gaussian equivalence holds and the deterministic component $\hat{\boldsymbol{F}}\boldsymbol{S}\boldsymbol{\kappa}_s$ vanishes at a rate of $\tilde{\mathcal{O}}(k^{-\delta})$ for some $\delta > 0$, then there exists a finite polynomial degree $p$ such that the polynomial activation $\hat{\sigma}_p(\cdot)$ (as defined in Theorem 4.12) yields equivalent generalization performance to that of the original nonlinear activation $\sigma(\cdot)$. In their proof, the vanishing nature of the deterministic term is used to show that $\hat{\sigma}_p(\cdot)$ and $\sigma(\cdot)$ induce the same first two conditional moments, thereby ensuring asymptotic equivalence in generalization (ICL) error.

Similarly, in our setting, the vanishing nature of the term $\hat{\boldsymbol{F}}\boldsymbol{S}\boldsymbol{\kappa}_s$—together with Assumption 4.8—implies the following bounds:

$$\left\| \mathbb{E}\left[ \sigma(\hat{\boldsymbol{F}}\text{vec}(\boldsymbol{H_Z})) \mid s, \boldsymbol{\kappa}_s \right] - \mathbb{E}\left[ \hat{\sigma}_p(\hat{\boldsymbol{F}}\text{vec}(\boldsymbol{H_Z})) \mid s, \boldsymbol{\kappa}_s \right] \right\|_2 = o(1), \tag{S34}$$

$$\left\| \mathbb{E}\left[ \sigma(\hat{\boldsymbol{F}}\text{vec}(\boldsymbol{H_Z}))\boldsymbol{o}^T \mid s, \boldsymbol{\kappa}_s \right] - \mathbb{E}\left[ \hat{\sigma}_p(\hat{\boldsymbol{F}}\text{vec}(\boldsymbol{H_Z}))\boldsymbol{o}^T \mid s, \boldsymbol{\kappa}_s \right] \right\|_F = o(1), \tag{S35}$$

$$\left\| \text{Cov}\left( \sigma(\hat{\boldsymbol{F}}\text{vec}(\boldsymbol{H_Z})) \mid s, \boldsymbol{\kappa}_s \right) - \text{Cov}\left( \hat{\sigma}_p(\hat{\boldsymbol{F}}\text{vec}(\boldsymbol{H_Z})) \mid s, \boldsymbol{\kappa}_s \right) \right\|_2 = o(1), \tag{S36}$$

where $\boldsymbol{o} := \boldsymbol{P}_\perp \text{vec}(\boldsymbol{H_Z})$. These bounds suffice to prove the equivalence between the original activation $\sigma(\cdot)$ and the polynomial approximation $\hat{\sigma}_p(\cdot)$, as explained by [13]. Therefore, it remains to verify that

$$|\hat{\boldsymbol{f}}_i^T \boldsymbol{S}\boldsymbol{\kappa}_s| = \tilde{\mathcal{O}}(k^{-\delta}) \quad \text{for all } i \in \{1, \ldots, k\}, \tag{S37}$$

for some $\delta > 0$, where $\hat{\boldsymbol{f}}_i$ denotes the $i$-th row of $\hat{\boldsymbol{F}}$.

To analyze $|\hat{\boldsymbol{f}}_i^T \boldsymbol{S}\boldsymbol{\kappa}_s|$, we begin by expanding the expression:

$$|\hat{\boldsymbol{f}}_i^T \boldsymbol{S}\boldsymbol{\kappa}_s| = \left| \hat{\boldsymbol{f}}_i^T \boldsymbol{S}(\boldsymbol{S}^T\boldsymbol{S})^{-1}\boldsymbol{S}^T\text{vec}(\boldsymbol{H_Z}) \right| \tag{S38}$$

$$\leq \left| \boldsymbol{f}_i^T \boldsymbol{S}(\boldsymbol{S}^T\boldsymbol{S})^{-1}\boldsymbol{S}^T\text{vec}(\boldsymbol{H_Z}) \right| + \eta \left| \boldsymbol{g}_i^T \boldsymbol{S}(\boldsymbol{S}^T\boldsymbol{S})^{-1}\boldsymbol{S}^T\text{vec}(\boldsymbol{H_Z}) \right|, \tag{S39}$$

where we used the decomposition $\hat{\boldsymbol{f}}_i = \boldsymbol{f}_i + \eta\boldsymbol{g}_i$, and $\boldsymbol{g}_i$ is the $i$-th row of the gradient matrix $\boldsymbol{G}$ defined in (7). The first term corresponds to the contribution from the randomly initialized feature matrix $\boldsymbol{F}$, while the second accounts for the effect of the gradient update.

Finally, invoking the gradient decomposition and the corresponding bounds from Appendix C, together with the fact that $\boldsymbol{P} = \boldsymbol{S}(\boldsymbol{S}^\top \boldsymbol{S})^{-1} \boldsymbol{S}^\top$ has rank $O(1)$ and Assumptions 4.4–4.5, we conclude that both terms in (S39) vanish at the desired rate. In particular, the scaling in Assumption 4.5, the low-rank structure of $\boldsymbol{P}$, and the spiked-covariance in Assumption 4.4 ensure that the contributions of the $\eta$-weighted gradient term and the spiked directions in $\mathrm{Cov}(\mathrm{vec}(\boldsymbol{H_Z}))$ remain controlled, which completes the proof.

In summary, our results indicate that although the output of the attention layer, $\mathrm{vec}(\boldsymbol{H_Z})$, exhibits a heavy-tailed distribution [41], the application of the random matrix $\boldsymbol{F}$ attenuates the heavy tails. Moreover, training the first-layer weights $\boldsymbol{F}$ with a single gradient step has a negligible effect on the analysis. Together, these insights allow us to transfer known results from supervised learning with two-layer neural networks to the in-context learning setting of Transformers with nonlinear MLPs. This connection opens promising avenues for analyzing complex, realistic models using Gaussian equivalence techniques.

## F  Extension to inputs with non-zero mean

In the main body of our proofs, we have assumed zero-mean inputs, i.e., $\boldsymbol{\mu}_{x,s} = \boldsymbol{0}$ for all data sources $s$, to streamline the exposition and focus on the core aspects of the Gaussian equivalence framework. However, in practical scenarios, input data often has non-zero mean, necessitating a generalization of our theoretical results to handle such cases.

The key observation enabling this extension is that the mean vector $\boldsymbol{\mu}_{x,s}$ of each Gaussian data source introduces a structured, low-rank perturbation in the representation of the attention output. Formally, we have:

$$\boldsymbol{H_Z} \mid s = (\boldsymbol{x}_{\ell+1} \mid s - \boldsymbol{\mu}_{x,s})\boldsymbol{b}^T + \boldsymbol{\mu}_{x,s}\boldsymbol{b}^T, \tag{S40}$$

which separates the stochastic (zero-mean) and deterministic (mean) components. Consequently, the vectorized attention output $\mathrm{vec}(\boldsymbol{H_Z})$ becomes non-zero-mean, but its non-central second moment still captures the combined statistical behavior of both components.

To rigorously extend our results to this more general setting, the following changes are required.

1. Substitution of covariances with second moments:
   - In all analytical steps where the covariances of $\boldsymbol{x}_i|s$ and $\mathrm{vec}(\boldsymbol{H_Z})$ appear, we need to replace it with the corresponding non-central second moment:

$$\begin{aligned}
\mathrm{Cov}(\mathrm{vec}(\boldsymbol{H_Z})) &\quad \rightsquigarrow \quad \mathbb{E}[\mathrm{vec}(\boldsymbol{H_Z})\mathrm{vec}(\boldsymbol{H_Z})^T], \\
\boldsymbol{\Sigma}_{x,s} = \mathrm{Cov}(\boldsymbol{x}_i|s) &\quad \rightsquigarrow \quad \mathbb{E}[\boldsymbol{x}_i\boldsymbol{x}_i^T|s].
\end{aligned}$$

   - This replacement maintains the validity of our results since the spike due to the mean is treated analogously to spikes in the spiked covariance model in Assumption 4.4.

2. Bounding mean-induced terms in gradient decomposition:
   - The primary technical adjustment is required in the proof of the gradient decomposition (Appendix C).
   - The decomposition must now include additional cross-terms involving $\boldsymbol{\mu}_{x,s}$, which must be carefully bounded using concentration inequalities and structural assumptions on the data distribution (e.g., bounded mean norm, low-rank behavior).
   - These bounds ensure that the mean-induced components do not asymptotically dominate or distort the gradient behavior established under the zero-mean assumption.

By implementing the above modifications—namely, using non-central second moments and bounding mean-induced gradient terms—we can extend our theoretical guarantees to the case of non-zero-mean Gaussian inputs. This generalization not only reinforces the robustness of our analysis but also broadens its relevance to real-world applications, where input means are rarely zero in practice.

# G  ICL errors per-source in the setting of Figure 2

For the sake of completeness, we illustrate the ICL errors per-source (corresponding to each source) in the setting of Figure 2. Mathematically, the per-source error is defined as $\mathbb{E}[(y_{\ell+1} - \hat{y})^2 | s = \hat{s}]$ where source indicator $\hat{s} \in \{0, 1\}$ since we consider settings with two different data sources. For all of the figures below, on the left, the ICL error corresponding to source 0, i.e., $\mathbb{E}[(y_{\ell+1} - \hat{y})^2 | s = 0]$, is plotted while the ICL error for source 1, i.e., $\mathbb{E}[(y_{\ell+1} - \hat{y})^2 | s = 1]$, is shown on the right. Figures 4, 5, and 6 illustrate the per-source ICL errors in the settings of Figure 2 (a), (b), and (c), respectively. Namely, Figure 4 shows the changes in the per-source ICL errors when varying the input covariance and Figure 5 displays the effects of changing the task covariance on the per-source ICL errors, while Figure 6 depicts the per-source ICL errors for different noise levels. In all of these cases, the first data source (source 0) is kept fixed while the aforementioned properties of the second data source are modified, as detailly explained in the caption of Figure 2. The results indicate two primary points. First, per-source ICL errors for the Transformer (approximately) match those of the equivalent model, indicating that the equivalence specified by Theorem 4.12 is useful for studying per-source ICL errors as well. Second, the trends for the ICL errors (which is the average of the per-source errors) in Figure 2 are also observed for each of the per-source errors, providing further evidence for our conclusions.

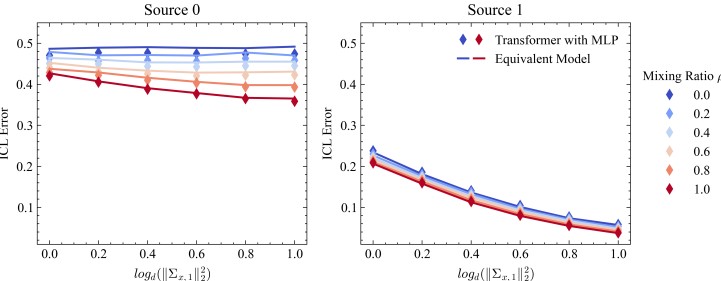

Figure 4: Per-source ICL errors in the case of Figure 2(a): impact of varying input covariance of source 1 on the per-source ICL errors.

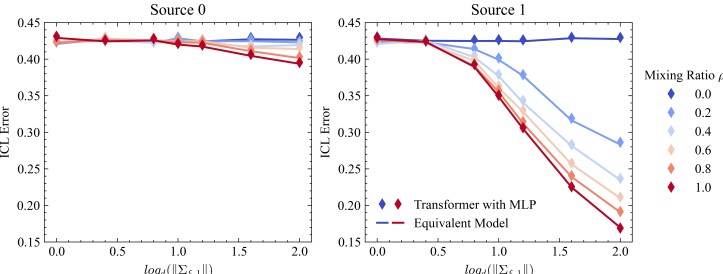

Figure 5: Per-source ICL errors in the case of Figure 2(b): effect of altering the task covariance of source 1 on the per-source ICL errors.

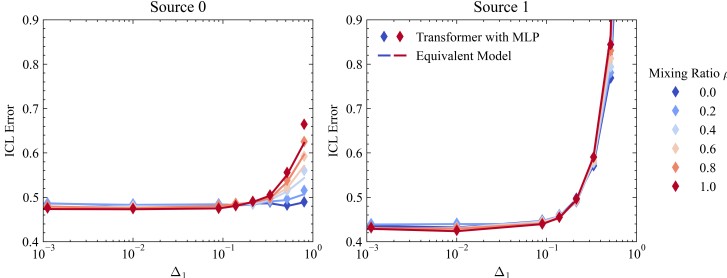

Figure 6: Per-source ICL errors in the case of Figure 2(c): result of changing noise level of source 1 on the per-source ICL errors.

