# OpenReview forum: "How Data Mixing Shapes In-Context Learning: Asymptotic Equivalence for Transformers with MLPs"
_NeurIPS.cc/2025/Conference — NeurIPS 2025 poster_

### Official Review · Reviewer_CMBv · 2025-07-01

**Clarity:** 3
**Significance:** 2
**Originality:** 2
**Rating:** 4
**Confidence:** 4

**Summary:**

The authors study in-context learning (ICL) for a *linear-attention* Transformer whose MLP head has two layers: the first is updated by **one gradient step**, the second is fitted.
Under joint high-dimensional scaling, they show (Theorem 4.12) that the model’s ICL error matches that of a suitably chosen degree-p polynomial predictor.

Empirically (Figures 1–3), they confirm that:

- The nonlinear MLP head beats the purely linear baseline.
- The polynomial surrogate tracks the MLP’s performance.

Their framework also enables analysis of “data-mixing” effects, showing that mixtures with **structured covariance** and **low noise** are most effective for ICL.

**Questions:**

See weaknesses.

**Ethical Concerns:**

["NO or VERY MINOR ethics concerns only"]

**Final Justification:**

The authors have convinced me of the distinction between their work and prior studies. Although this work does not fully address the mechanism of in-context learning (ICL), it provides valuable insights in simplified settings.

**Limitations:**

Yes.

**Quality:**

2

**Strengths And Weaknesses:**

Strengths:

1. The paper is well-written and clear.
2. The paper is addressing an important problem of the role of MLP in in context learning.
3. The assumptions are stated clearly.

Weaknesses

1. The paper only considers *linear attention*, which limits its generality. Prior work has shown that for any statistical ICL task, there exists a fixed **non-linear attention** mechanism that achieves Bayes error as the context length tends to infinity, an assumption used in this paper. (Section 5.1 of this paper [^1] )

2. The same work[^1] provides shows evidence that how the MLP enables ICL in one-layer Transformers with any nonlinear attention head. The current paper does not cite this or contrast its contributions, weakening its novelty and context.

3. Both the current paper and prior work[^1] restrict their analysis to one-layer Transformers. However, empirical studies show a significant performance leap from one to two layers, with diminishing returns improvement  beyond two layer. Understanding how ICL emerges across **multiple layers** is crucial, and ignoring this makes the theoretical claims less applicable to real-world models.

---

[^1]: Abedsoltan, A., Radhakrishnan, A., Wu, J., & Belkin, M. (2024). *Context-Scaling versus Task-Scaling in In-Context Learning*. [arXiv:2410.12783](https://arxiv.org/abs/2410.12783)

---

> ### Author Rebuttal · Authors · 2025-07-29
>
> Thank you for your detailed and constructive feedback. We appreciate your recognition of the paper’s clarity, relevance to ICL, and well-stated assumptions. Below we address your concerns and outline how the revised version will incorporate the suggested clarifications and citations.
>
> > 1. The paper only considers linear attention, which limits its generality. Prior work has shown that for any statistical ICL task, there exists a fixed non-linear attention mechanism that achieves Bayes error as the context length tends to infinity, an assumption used in this paper. (Section 5.1 of this paper [^1])
>
> We agree that nonlinear attention plays a critical role in real-world Transformers. Our choice to focus on trainable linear attention is intentional, as it enables rigorous high-dimensional asymptotic analysis, which is currently out of reach for nonlinear attention mechanisms. This analytical tractability allows us to isolate the role of the nonlinear MLP head and precisely quantify its impact on ICL performance. That said, recent empirical work (Han et al., NeurIPS 2024) suggests that carefully designed linear attention variants can approach softmax performance—supporting the continued relevance of the linear setting for both analysis and applications.
>
> *(Han et al., NeurIPS 2024): Bridging the Divide: Reconsidering Softmax and Linear Attention*
>
> > 2. The same work [^1] provides shows evidence that how the MLP enables ICL in one-layer Transformers with any nonlinear attention head. The current paper does not cite this or contrast its contributions, weakening its novelty and context.
>
> We thank the reviewer for pointing out this relevant work (Abedsoltan et al., 2024). We will cite it in the revised version and highlight the following contrasts. Specifically, Abedsoltan et al. (2024) study a simplified Transformer architecture with a fixed (non-trainable) nonlinear attention mechanism—omitting key, query, and value weights—and an optional MLP head. They show that this setup converges to a optimal (consistent) estimator in the infinite-context limit via a Hilbert estimate. In contrast, our work investigates the role of a nonlinear MLP head in a Transformer with trainable linear attention, under a data mixing setting. We rigorously show that the ICL error of this model asymptotically matches that of a finite-degree polynomial predictor—thus providing a new theoretical understanding of how nonlinear MLPs enable feature learning and improve performance over linear baselines (as shown in Figure 1). Importantly, we also analyze how these gains depend on the structure of data mixing and training of the MLP.
> While both papers explore MLPs in the context of ICL, our contributions are complementary and novel in several key ways:
>
> - Our analysis explicitly models heterogeneous task mixtures with structured covariances, revealing how data mixing governs feature learning—an aspect not studied in Abedsoltan et al. (2024).
>
> - We derive an asymptotic equivalence between Transformer predictors and finite-degree polynomial models under high-dimensional scaling, providing quantitative performance predictions.
>
> - Our empirical results (Figures 1–3) validate this theoretical correspondence, demonstrating that nonlinear MLP heads consistently outperform linear ones in alignment with the predicted trends.
>
> To further clarify the distinction, we present a detailed comparison between our work and the cited study in the following table:
>
> |                  | Abedsoltan et al. (2024)                                               | *Ours*                                                                                    |
> |------------------|------------------------------------------------------------------------|-------------------------------------------------------------------------------------------|
> | **Attention**    | **Fixed (non-trainable)** *non-linear* attention  feature map (Q=K=V=I; row‑wise $\ell_1$‑normalized kernel smoother) with an optional MLP head after it      | **Trainable** *linear* attention with an nonlinear MLP after the attention                        |
> | **Data model**   | **Homogeneous** tasks (linear regression, teacher-student, decision trees) | **Heterogeneous** task mixtures with *multiple* data sources (nonlinear regression sources, with structured data, task, and noise) |
> | **Theory**       | **Linking attention to kernel smoother**: Consistency (optimal in the infinite‑context limit) via the Hilbert estimate   | **Rigorous high-dimensional analysis**: asymptotic equivalence to a polynomial model   |
> | **New insights** | Suitably chosen non‑linear attention can achieve optimal ICL in the infinite‑context limit          | Precisely how MLPs in Transformers drive ICL and when feature learning emerges from *data-mixing*         |
>
>
> > 3. Both the current paper and prior work[^1] restrict their analysis to one-layer Transformers. However, empirical studies show a significant performance leap from one to two layers, with diminishing returns improvement beyond two layer. Understanding how ICL emerges across multiple layers is crucial, and ignoring this makes the theoretical claims less applicable to real-world models.
>
> We agree that multi-layer Transformers are highly relevant, and we appreciate the suggestion. Our choice to analyze a one-layer model (involving linear attention and a two-layer neural network) is consistent with much of the foundational ICL theory (e.g., [23, 41, 37]) and enables precise analytical results that are not yet tractable in deeper models. The one-layer case is not only analytically clean but also sheds light on where and how ICL behavior emerges—in this case, via MLP-driven feature learning under data mixing.
> We will clearly state this as a limitation and a natural direction for future work. We believe our theoretical framework forms a rigorous starting point for eventual generalizations to deeper architectures.
>
> ---
>
> In summary, we thank the reviewer for highlighting important directions and related work. We will revise the paper to:
> * Cite and clearly differentiate from Abedsoltan et al. (2024),
> * - Clarifying the novelty of our polynomial surrogate theory under high-dimensional scaling,
> * - Highlighting our unique analysis of data mixing and the role of nonlinear MLPs in Transformers,
> * Explicitly discuss the one-layer limitation and suggest paths to extend our work.
>
> We hope these clarifications will positively impact your evaluation. We believe our contributions offer new theoretical tools and insights into ICL with MLPs—complementary to prior work and of relevance to both theory and practice.
>
>
> *[^1] (Abedsoltan et al., 2024): Context-Scaling versus Task-Scaling in In-Context Learning, arXiv:2410.12783.*

---

> > ### Comment · Reviewer_CMBv · 2025-08-05
> >
> > Thank you to the authors for the clarification. I still believe it is important to consider more than one layer to meaningfully understand the mechanism of in-context learning (ICL). That said, I appreciate the authors’ effort to isolate and study the role of MLPs in ICL. While prior work has provided insights into the role of MLPs—particularly in settings where the attention heads lack learnable key, query, and value parameters—this work takes a different angle. I acknowledge this distinction and will increase my score accordingly.

---

> > > ### Author Response · Authors · 2025-08-07
> > >
> > > Thanks for your response and raising your rating. We appreciate your recognition of the distinction from the prior work.

---

### Official Review · Reviewer_Z4dn · 2025-07-01

**Clarity:** 3
**Significance:** 1
**Originality:** 2
**Rating:** 2
**Confidence:** 3

**Summary:**

The paper consists of theoretical and empirical parts. In the theoretical part, they consider 1 layer transformer (linear self-attention+2 layer MLP with non-linear activation) (equivalently written as in Equation (5)) where (equivalent representation of) all self-attention + first MLP layer parameters are trained with a single gradient descent step, and after that, second layer in MLP is taken to be ridge regression loss minimizer on separate data. The main theoretical result is that this transformer is asymptotically (Assumption 4.1) equivalent to structured polynomial predictor (Equation 13), when observing ICL error on ICL non-linear regression data (from mixture of sources). Empirically, they confirm this equivalence by simulations, and experiment with the impact of model size and data characteristics to successfully learned ICL.

**Questions:**

1. Can you elaborate on meaning of equation 13? Is it not the same as Equation (5) but with a polynomial activation (and updated F)? What is the benefit (apart maybe theoretical convenience) of showing that we can replace activation function with a polynomial?
2. In the sentence on lines 130-131, there is no mention of transformers being able to capture non-linearities in data with a non-linearity on self-attention (such as usually used softmax). Can you say why, or mention it in this sentence?
3. Can you justify more the focus on the average task error across all sources in the evaluation (i.e. the introduced ICL Error)? This makes the train and test distribution different. Instead, I would perhaps consider per source error (which is also tractable in the setting of the paper – 2 sources). Have the authors looked into this already?
4. In line 168 (potentially elsewhere as well), it is not specified which matrix norm is being used?
5. On line 282, where does this intuition come from?
6. Have you tried having more than 2 data sources in the simulations?

**Ethical Concerns:**

["NO or VERY MINOR ethics concerns only"]

**Final Justification:**

I appreciate the authors' engagement, but my concerns still persist, as noted in my response to the rebuttal. I will retain my current rating, though I understand that it is lower than the other reviewers' scores. I will not "fight" to reject this paper, but would like to keep my opinion expressed (hence not changing the score).

**Limitations:**

There is a limitation introduced with the training procedure of the model. It isn’t clear to me how this limitation would  be solved if we turn to training a transformer with GD or Adam.

**Paper Formatting Concerns:**

Line 136, fix indexing i and j

Equation 8, remove commas

**Quality:**

3

**Strengths And Weaknesses:**

## Strengths:

The theoretical paper has nicely set notation, well explained data and model setup. The theoretical assumptions and results are clearly stated. They follow some nicely cited previous work. The experiments are also clear.

## Weaknesses:

The goal of the paper would be to bridge the theoretical setup closer to the practical use cases of transformers, by incorporating nonlinear MLP after linear self-attention. Apart from the theoretical assumptions in section 4.1, there is also the somewhat weird training procedure of the MLP. Namely, training one linear layer with only one step of GD and fitting the other with ridge regression is also very far from practice, but this divergence is not discussed.

I'm not sure of the significance of the main theoretical result (Theorem 4.12) is. This may be my misunderstanding, but it would be good for the authors to comment on this more/discuss this in the paper. Namely, how this understanding may go beyond existing/classical intuition (which is what the current Conclusion reads as).

Similarly, the empirical part seems to lack a significant/original conclusion in my opinion:
- Figure (1) isn’t so surprising to me, namely conclusion on line 256 (nonlinear > linear)
- In Figure 2a), I don’t agree that the conclusion is that a more structured covariance matrix leads to more ICL (line 269). It can happen that ICL on the first (easier) source is easy to learn and the model has acquired that ability. In addition, ICL on structured data (second source) is harder, and the model learns it only with more examples from that source. Then the decrease in the ICL error would be justified (when increasing $\rho$), but it wouldn’t be the case that more structured data leads to better ICL. Another reason to believe this is the case is to observe the $\rho=0.6, 0.8$ cases: The ICL error decreases and then increases again as structure is increasing. I suspect that plotting the error per source may reveal that the error on the first source is slightly increasing, while on the second source is decreasing a lot—together making the U-shaped curve.
- Conclusion on line 273 is also not surprising to me (lower noise leading to performance gains).
- I’d plot some more, higher values than 2 in Figure 3b) to check that the claimed trend of decreasing is true.

I would explain Equation (5) in more detail, e.g. It is not stated that $\Gamma$ and the first linear layer in MLP are incorporated in $F$.

---

> ### Author Rebuttal · Authors · 2025-07-29
>
> Thank you for your detailed review and the thoughtful feedback. We appreciate your recognition of the clarity in our theoretical exposition, assumptions, and experimental design. Below, we respond to your concerns and questions point-by-point:
>
> > The goal of the paper would be to bridge the theoretical setup closer to the practical use cases of transformers, by incorporating nonlinear MLP after linear self-attention. Apart from the theoretical assumptions in section 4.1, there is also the somewhat weird training procedure of the MLP. Namely, training one linear layer with only one step of GD and fitting the other with ridge regression is also very far from practice, but this divergence is not discussed.
>
> We agree that our training procedure differs from standard end-to-end training pipelines. However, this two-phase scheme—gradient descent on the first layer followed by ridge regression on the second—is well-motivated by prior theoretical work on MLPs [3, 9]. It can be interpreted as an idealized form of a two-phase learning schedule with layer-wise learning rate separation: a large learning rate applied briefly to early layers, followed by fine-tuning of later layers. This separation enables tractable analysis while preserving meaningful feature learning dynamics. We will expand on this interpretation in the revised paper.
>
> > I'm not sure of the significance of the main theoretical result (Theorem 4.12) is. This may be my misunderstanding, but it would be good for the authors to comment on this more/discuss this in the paper. Namely, how this understanding may go beyond existing/classical intuition (which is what the current Conclusion reads as).
>
> > Can you elaborate on meaning of equation 13? Is it not the same as Equation (5) but with a polynomial activation (and updated F)? What is the benefit (apart maybe theoretical convenience) of showing that we can replace activation function with a polynomial?
>
> Thank you for raising this important point. Theorem 4.12 establishes an asymptotic equivalence between the ICL behavior of a Transformer with an MLP and that of a polynomial model. This result is significant for several reasons:
>
> 1. **Analytical tractability**: Equation (13) admits a much simpler analysis, allowing precise characterization of ICL error behavior of the Transformer under data mixing and feature learning.
> 2. **Interpretability**: It sheds light on the function class learned by the Transformer—specifically, that it effectively learns a low-degree polynomial approximation of the task function.
> 3. **Nonlinearity design**: This equivalence opens the door to optimizing nonlinearities in MLPs through polynomial surrogate analysis.
>
> These insights extend beyond classical intuition and align with recent trends in Gaussian equivalence literature [e.g., 13, 18]. We appreciate the suggestion and will clarify the conceptual impact of this result in the camera-ready version.
>
>
> > Similarly, the empirical part seems to lack a significant/original conclusion in my opinion:
>
> We respectfully disagree with you in this regard. While some findings may align with prior intuitions, we believe our contributions go beyond merely confirming them. For example, the relationship among in-context learning, task covariance, and feature learning (illustrated in Figures 2b and 3b) is quite interesting and original.
>
> > - Figure (1) isn’t so surprising to me, namely conclusion on line 256 (nonlinear $>$ linear)
>
> The observation in Figure 1 is meaningful because nonlinearity alone does not guarantee better ICL performance. As shown in prior work (e.g., [18]), nonlinear models can underperform linear models if feature learning is ineffective. Our result demonstrates that even one-step feature learning enables the Transformer to outperform its linear counterpart—highlighting the utility of the MLP layer in ICL.
>
> > - In Figure 2a), I don’t agree that the conclusion is that a more structured covariance matrix leads to more ICL (line 269). $\cdots$ Then the decrease in the ICL error would be justified (when increasing $\rho$), but it wouldn’t be the case that more structured data leads to better ICL. $\cdots$
>
> Regarding Figure 2a and line 269: we clarify that our conclusion is **not** that more structured covariance *always* leads to better ICL. Rather, increasing the **proportion** of structured data (denoted with $\rho$) improves ICL. This aligns with your interpretation, and we will reword the corresponding sentence to avoid misreading.
>
> > - Conclusion on line 273 is also not surprising to me (lower noise leading to performance gains).
>
> While the conclusion that lower noise leads to better performance may seem unsurprising in a standard single-source setting, our focus is on how noise interacts with data mixing in multi-source scenarios. In this context, analyzing the role of noise becomes nontrivial and offers valuable insights into how noise levels influence the effectiveness of different mixing strategies—an aspect that, to our knowledge, has not been explicitly studied before.
>
> > - I’d plot some more, higher values than 2 in Figure 3b) to check that the claimed trend of decreasing is true.
>
> For Figure 3b: we have indeed run simulations for larger values of $\log_d(\eta) > 2$ and found that the decreasing trend in ICL error continues, confirming the conclusion presented.
>
> > I would explain Equation (5) in more detail, e.g. It is not stated that $\Gamma$ and the first linear layer in MLP are incorporated in $F$.
>
> Thank you for pointing this out. In response to your feedback, we will revise the explanation around Equation (5) to explicitly clarify that both $\Gamma$ and the first linear layer of the MLP are incorporated into the definition of $F$.
>
> > In the sentence on lines 130-131, there is no mention of transformers being able to capture non-linearities in data with a non-linearity on self-attention (such as usually used softmax). Can you say why, or mention it in this sentence?
>
> As explained in the original Transformer paper (Vaswani et al., 2017), softmax (the nonlinearity in the attention) is primarily used to compute normalized weights for value aggregation. While this also makes the model nonlinear, we isolate and study the role of nonlinear MLPs in Transformers for capturing task-related nonlinearity by using linear attention. We will add a clarification in the relevant sentence.
>
>
> > Can you justify more the focus on the average task error across all sources in the evaluation (i.e. the introduced ICL Error)? This makes the train and test distribution different. Instead, I would perhaps consider per source error (which is also tractable in the setting of the paper – 2 sources). Have the authors looked into this already?
>
> Great question. We have examined per-source ICL error and found that it also behaves in line with our theoretical predictions. However, we focus on the average ICL error to:
>
> 1. Have a single evaluation criterion, capturing the overall ICL generalization capacity across mixed-task distributions.
> 2. Use a fixed evaluation criterion independent of mixing weights, making it easier to interpret performance across different scenarios.
>
> We will clarify this rationale in the revised version and may include additional plots in the appendix showing per-source behavior.
>
> > In line 168 (potentially elsewhere as well), it is not specified which matrix norm is being used?
>
> As noted in the appendix (Notation section), $\|\cdot\|$ denotes the spectral norm and $\|\cdot\|_F$ the Frobenius norm. We will move the notation section to the main text if space permits.
>
>
> > On line 282, where does this intuition come from?
>
> The intuition is derived from the fact that feature learning enables the model to align with task vectors $\boldsymbol{\xi}$, which can be shown via an analysis of one‑step gradient (e.g., see Section 3.2 of reference [3] for a detailed explanation of this behavior in the case of supervised learning with MLPs). Therefore, when task vectors are isotropic (as in Figure 3a), there is little structure to exploit, and feature learning becomes ineffective—resulting in flat ICL error curves.
>
>
> > Have you tried having more than 2 data sources in the simulations?
>
> Yes, we have experimented with 3 or more sources and observed similar trends. We chose the 2-source setup for clearer visualization and more direct interpretation of mixing effects. If needed, we are happy to share additional results.
>
>
> > Line 136, fix indexing i and j. Equation 8, remove commas.
>
> Thank you for these suggestions. We will revise the text and correct the noted typos accordingly.
>
> ---
>
> Overall, we believe our responses address all the concerns raised in your review. We hope that these clarifications, along with your acknowledgment of the paper’s strengths, lead to a reconsideration of your evaluation and ultimately, an increase in your rating. We appreciate your time and thoughtful evaluation.

---

> > ### Author Response · Authors · 2025-08-07
> >
> > Dear Reviewer Z4dn,
> >
> > Although the author-reviewer discussion period is drawing to a close, we have not yet received any response from you following our rebuttal. We believe your participation in the discussion is important for a fair evaluation of our work. Please let us know if you have any remaining questions or concerns after reading our rebuttal.

---

> > ### Comment · Reviewer_Z4dn · 2025-08-08
> >
> > Thank you for the detailed answers and further clarifications! Below I discuss my updated views and explain my thoughts that influenced the rating.
> >
> > > We agree that our training procedure differs from standard end-to-end training pipelines.
> >
> > Thanks for clarifying! This work is promoted as bridging the gaps from prior work, and stated that prior work has “limited relevance to realistic settings”; but prior work such as [37,15] doesn’t have this assumption in training, but rather uses a more realistic training with Adam. It is necessary to think about the tradeoff of bridging a limitation of prior work (no MLPs) but introducing a new limitation (training being different, reparametrization of model used (Equation 5)).
> >
> > What would help justify these new limitations introduced is if there were experiments with **transformers actually trained with Adam**; the paper only discusses transformers trained with this unusual procedure.
> >
> > >  Theorem 4.12 establishes an asymptotic equivalence between the ICL behavior of a Transformer with an MLP and that of a polynomial model.
> >
> > But this polynomial model is exactly a reparametrization of the linear self-attention+MLP with polynomial activation function (and the {self-attention weights+first layer of MLP} being updated by one step of GD). The main conclusion seems to be that asymptotically we can approximate non-linearity in MLP with a polynomial function?
> >
> > > The observation in Figure 1 is meaningful because nonlinearity alone does not guarantee better ICL performance. As shown in prior work (e.g., [18]), nonlinear models can underperform linear models if feature learning is ineffective.
> >
> > There have been cases when linear models outperform non-linear but because the task itself is linear (see [37]). So while the conclusion (i) on line 256 is true, I believe it is important to emphasize that it is likely to be highly dependent on the task, and has been verified only for the task considered in the paper.
> >
> > Additionally, it isn’t clear that the same conclusion would hold if the models were trained with Adam or SGD. The current conclusion on line 256 should hence read: after the first step of feature learning, linear self-attention with MLP outperforms the counterpart without MLP.
> >
> > > Regarding Figure 2a and line 269: we clarify that our conclusion is **not** that more structured covariance always leads to better ICL. Rather, increasing the **proportion** of structured data  improves ICL.
> >
> > Could the authors please restate this in the paper. For example, the current abstract reads ‘we identify **key properties of high-quality data sources** (low noise, **structured covariances**)’. This seems to say more structured covariance=better ICL.
> >
> > I still suspect that what happens is that the first source is easier, needing less examples, when the second (structured one) is harder and needs more examples to be learned. So increasing the proportion of samples from structured source will (trivially) improve the ICL error **on that source only** and thus also the total ICL error. However, this makes the conclusion unsurprising: if one source is harder to learn, providing more samples from that source is always beneficial and improves the error.
> >
> > To test whether this is the case, observe ICL error on the sources **separately**. If the ICL error on the first source is flat or slowly increasing as $\rho$ gets larger, and on the second source ICL error is decreasing, then my hypothesis is true. Can the authors confirm or deny whether this happens?
> >
> > > ... lower noise leads to better performance may seem unsurprising ... (this work uncovers) how noise levels influence the effectiveness of different mixing strategies
> >
> > The conclusion in lines 273-274 doesn’t seem to reflect on how noise levels influence the effectiveness of different mixing strategies.
> >
> > Again, this plot seems intuitively explainable by decoupling it to simpler and previously verified statements: 1) higher noise leads to worse performance (on second source), 2) higher proportion of second source data leads to better performance. Thus, I continue to struggle to see the significance.
> >
> > > ... we will revise the explanation around Equation (5) …
> >
> > I appreciate the confirmation that the authors will clarify Equation (5). Additionally, it is important that authors correct the misleading statements in the paper around this. For example, the abstract states ‘... the MLP comprises two layers, with **the first trained via a single gradient step** ...’ A reader would get the impression that only the first MLP layer is trained with one step of GD, where all the formulas suggest that 1) self-attention weights+first MLP layer are reparametrized to a single linear layer $F$ and 2) 1 step of GD is performed on this linear layer $F$.
> >
> > I’d like to thank the authors again for their responses and useful clarifications. I took all the responses into account in the final evaluation.

---

> ### Author Response · Authors · 2025-08-09
>
> We thank the reviewer for the detailed response. As we received it during the final 14 hours of the discussion period, we provide concise clarifications and outline our position as follows.
>
> ### Regarding the training procedure
>
> > $\cdots$ prior work such as [37,15] doesn’t have this assumption in training, but rather uses a more realistic training with Adam
>
> The results in [37] and [15] address different questions from ours. Specifically, our work provides an **asymptotic equivalence result** that enables precise performance characterization, whereas [15] studies the in-context learnability of a certain function class, and [37] relates the mechanism of ICL to gradient descent. Because the nature of our result is different, our analysis requires an **analytically tractable training procedure** to establish the equivalence between the two models. Such **simplified training procedures** are common in the theoretical literature [3, 9, 10, 13], and are valuable when they capture the relevant effects of training. In our case, one-step gradient descent—used in many prior works [3, 9, 10, 13]—is sufficient for meaningful feature learning, as illustrated in Fig. 3.
>
>
> ### Regarding the main theoretical result
> > $\cdots$ The main conclusion seems to be that asymptotically we can approximate non-linearity in MLP with a polynomial function?
>
> Our theorem establishes that, when trained under the specified setting, the two models—a Transformer and its corresponding polynomial model—achieve **equivalent performance in terms of ICL error** (analogous to generalization error in this context) in the asymptotic limit. This result goes **well beyond basic function approximation**, as it shows equivalence in learning performance after training, rather than merely expressing a nonlinearity as a polynomial.
>
> ### Regarding the experimental results
>
> > There have been cases when linear models outperform non-linear but because the task itself is linear (see [37]). So while the conclusion (i) on line 256 is true, I believe it is important to emphasize that it is likely to be highly dependent on the task, and has been verified only for the task considered in the paper.
>
> We acknowledge this point and have already noted it when motivating the nonlinear setting on line 99. We will further emphasize this limitation in the discussion of the experimental results to clarify that the finding is task-dependent and verified only for the setting studied in this work.
>
> > Additionally, it isn’t clear that the same conclusion would hold if the models were trained with Adam or SGD. The current conclusion on line 256 should hence read: after the first step of feature learning, linear self-attention with MLP outperforms the counterpart without MLP.
>
> Our discussion in the paper focuses on the specified training setting. However, we expect the conclusion to also hold for models fully trained with optimizers such as Adam or SGD, since full training typically improves model performance.
>
> > I still suspect that what happens is that the first source is easier, needing less examples, when the second (structured one) is harder and needs more examples to be learned. So increasing the proportion of samples from structured source will (trivially) improve the ICL error on that source only and thus also the total ICL error. However, this makes the conclusion unsurprising: if one source is harder to learn, providing more samples from that source is always beneficial and improves the error.
>
> We clarify that, in our experiments, the **structured source is in fact easier to learn**, yielding lower error compared to the isotropic source.
>
> > To test whether this is the case, observe ICL error on the sources separately.
>
> As suggested, we will include the per-source ICL errors in the final version of the paper.
>
> > Again, this plot seems intuitively explainable by decoupling it to simpler and previously verified statements: 1) higher noise leads to worse performance (on second source), 2) higher proportion of second source data leads to better performance. Thus, I continue to struggle to see the significance.
>
> We interpret this comment as suggesting that adjusting the mixing ratio will always decrease only one of the per-source errors. However, this need not hold. For example, if two data sources have different noise levels for the same or related task, **improving performance on one source can also yield gains on the other**. Thus, the effects of data mixing can be more nuanced than the simplified view implies.
>
> ### Regarding the clarity
>
> We will further improve the paper’s clarity in line with the reviewer’s feedback.
>
> ---
>
> Overall, we thank the reviewer for their time and effort in reviewing our work and hope these clarifications are helpful.

---

### Official Review · Reviewer_G3y6 · 2025-07-03

**Clarity:** 3
**Significance:** 3
**Originality:** 4
**Rating:** 5
**Confidence:** 3

**Summary:**

This paper presents a theoretical analysis on how Transformers with nonlinear MLP heads perform in-context learning when trained on heterogeneous data. The authors establish a tractable analytical framework by proving an asymptotic equivalence between complex models and simpler polynomial predictors. This result is achieved through an application of Gaussian universality theory and Hermite expansions under high-dimensional, proportional-limit asymptotics. The function surrogate enable derive closed-form expressions for ICL error, offering novel insights into how data mixing weights, noise levels, and covariance structures collectively determine learning outcomes.

**Questions:**

1. How feasible to extent the analysis to nonlinear attention, perhaps via kernelized approximation?
2. How do the dynamics of the polynomial surrogate change empirically after multiple gradient steps?
3. Could the data mixing heuristics be validated on a real-world benchmark, such as a multilingual corpus where mixing weights control language fractions?

**Ethical Concerns:**

["NO or VERY MINOR ethics concerns only"]

**Limitations:**

The analysis is limited to regression tasks, leaving its applicability to classification, or language modeling tasks unclear

**Quality:**

3

**Strengths And Weaknesses:**

**Strengths**
- This work fills a long-standing gap in the literature by incorporating both nonlinear MLP components and multi-source data mixing—two ubiquitous features of real-world models that prior linear-only analyses could not address.
- The analysis is rigorous and elegant. The authors skillfully coupling Gaussian universality with a rank-one decomposition of the gradient to analyze the feature learning dynamics after a single update step in first MLP layer.
- The results are insightful. By quantifying how factors like mixing weights, noise levels, and covariance structure shape ICL performance, the work provides clear data curation heuristics—for instance, prioritizing low-noise, structured covariances data sources.
- These theoretical predictions are validated through simulations that match the theory.

**Weaknesses**
- The linear-attention simplification limits the practical applications. The nonlinear attention mechanism is fundamental to the success of modern Transformer. There is no evidence or discussion on how feasible the analysis can be transfer.
- Real feature learning requires multiple optimization steps, and the dynamics likely change as optimization progress. The current analysis rely on one-step gradient on the first MLP layer is too restrictive.
- Requiring attention output covariances to have identical traces across all tasks and admit low-rank decompositions seems unlikely to hold in practice for natural language or vision contexts. The authors note this but do not quantify robustness.
- The motivation of assumption 4.5 is unclear and not argued.
- The experiments are exclusively rely on synthetic Gaussian data without testing on real benchmarks.

---

> ### Author Rebuttal · Authors · 2025-07-29
>
> We sincerely thank the reviewer for the thoughtful and thorough evaluation. We are particularly grateful that you recognized the novelty and rigor of our approach, the integration of nonlinear MLPs and data mixing, and the empirical alignment with our theoretical predictions. Below, we address each of the concerns and questions raised:
>
> > The linear-attention simplification limits the practical applications. The nonlinear attention mechanism is fundamental to the success of modern Transformer. There is no evidence or discussion on how feasible the analysis can be transfer.
>
> >  How feasible to extent the analysis to nonlinear attention, perhaps via kernelized approximation?
>
> We acknowledge that the use of linear attention is a simplifying assumption and agree that extending the analysis to nonlinear (e.g., softmax) attention is a valuable direction. We chose linear attention to preserve analytical tractability, which enabled our novel integration of nonlinear MLP heads and data mixing analysis. That said, recent empirical work (Han et al., NeurIPS 2024) suggests that carefully designed linear attention variants can approach softmax performance—supporting the continued relevance of the linear setting for both analysis and applications.
>
> Your suggestion to explore kernelized approximations for nonlinear attention is especially insightful. We agree this is a promising path for extending theoretical tools to richer architectures, though it introduces additional technical challenges and remains open for future work.
>
> *(Han et al., NeurIPS 2024): Bridging the Divide: Reconsidering Softmax and Linear Attention*
>
> > Real feature learning requires multiple optimization steps, and the dynamics likely change as optimization progress. The current analysis rely on one-step gradient on the first MLP layer is too restrictive.
>
> > How do the dynamics of the polynomial surrogate change empirically after multiple gradient steps?
>
> This is an important point. Our analysis focuses on a single (but large) gradient step to allow rigorous characterization of the feature dynamics, similar in spirit to prior work analyzing supervised learning with MLP settings (e.g., references [3, 9]). Empirically, we observe that this one-step update captures key aspects of feature learning and already yields strong alignment with theory. Extensions to multiple steps are indeed plausible—in fact, under appropriately scaled step sizes, our framework can accommodate a small number of steps without substantial analytical disruption. Nonetheless, a full multi-step theory, especially with nonlinearities, remains an exciting and non-trivial direction for future work—even for simpler MLP-only settings (see, e.g., reference [13]).
>
> > Requiring attention output covariances to have identical traces across all tasks and admit low-rank decompositions seems unlikely to hold in practice for natural language or vision contexts. The authors note this but do not quantify robustness.
>
> We appreciate this concern. The assumption of identical trace and low-rank structure is indeed primarily for analytical convenience, helping us isolate the core dynamics of interest. However, our experiments suggest that the predictions remain qualitatively robust even when this assumption is mildly violated. We will explicitly clarify this point in the revised manuscript and highlight the robustness behavior under partial relaxation.
>
> > The motivation of assumption 4.5 is unclear and not argued.
>
> Thank you for pointing this out. Assumption 4.5 controls the relationship between the step size and the covariance norm, ensuring that the feature update is nontrivial but remains within the regime where our asymptotic analysis is valid. While we briefly mention this in the current version, we agree that the rationale can be better justified. We will revise the manuscript to clearly articulate the role, motivation, and practical relevance of this assumption.
>
> > The experiments are exclusively rely on synthetic Gaussian data without testing on real benchmarks.
>
> > Could the data mixing heuristics be validated on a real-world benchmark, such as a multilingual corpus where mixing weights control language fractions?
>
> > The analysis is limited to regression tasks, leaving its applicability to classification, or language modeling tasks unclear
>
> These are all excellent suggestions. We focused on synthetic Gaussian settings to enable precise theoretical modeling and tight control over key variables like mixing weights, noise, and covariance. That said, we agree that empirical validation on real-world problems—such as language modeling tasks with multilingual corpora where mixing is naturally interpretable—is an important next step. We will include a discussion of this in the revised manuscript as a clear future direction. Finally, we would like to note that while our presentation focused on regression tasks for simplicity, our theoretical results trivially apply to classification (if the target/label function is chosen appropriately) as well. We will mention this point in the revised paper.
>
> ---
>
> Overall, we are grateful for your positive review and constructive suggestions. Your feedback has already helped improve the clarity and potential impact of our work, and we look forward to incorporating these insights into the final version.

---

> > ### Comment · Reviewer_G3y6 · 2025-08-04
> > **Thanks**
> >
> > I appreciate the time the authors have taken to address my concerns.
> >
> > However, my primary remain concern is - the paper lacks a bridge to real-world tasks. This tempers the work's immediate significance, and a promise to mention this as "future work" does not change the contribution of the paper as it stands today.

---

> > > ### Author Response · Authors · 2025-08-05
> > >
> > > Thanks for your response (and again for your positive review). We understand your concern about the connection to real-world tasks and agree that bridging this gap is an important direction. While our current work focuses on foundational theoretical insights derived from an asymptotic characterization of the ICL error (via equivalence) under controlled data mixing settings, we believe it lays important theoretical groundwork for future studies involving real-world tasks. We are grateful for your feedback and hope our clarifications have helped frame the scope and contributions more clearly.

---

> ### Author Response · Authors · 2025-08-07
>
> Dear reviewer G3y6,
>
> Since you noted that “my primary remaining concern is - the paper lacks a bridge to real-world tasks,” we conducted a new experiment specifically designed to address this point. Building on your suggestion in Question 3 of your review, we explored a real-world scenario involving multilingual sentiment analysis using the Multilingual Amazon Reviews Corpus (Keung et al., EMNLP 2020). This dataset contains customer reviews (with text and star ratings) in multiple languages, such as English and German. By treating English and German reviews as two distinct data sources, we can vary the mixing ratio across languages—mirroring the setup you proposed—and evaluate our framework in a more realistic setting.
>
> ---
>
> **New results on real-world tasks (multilingual sentiment analysis):**
>
> The ICL errors of both the Transformer and the equivalent model are shown for various GD step sizes and mixing ratios in the table below. Here, we define English reviews as source 0 and German reviews as source 1, so a mixing ratio of $\rho = 0.0$ corresponds to entirely English data, and $\rho = 1.0$ to entirely German data.
>
> | mixing ratio (model) / scale of GD step size | $log_d(\eta) = 0.0$ | $log_d(\eta) = 0.5$ | $log_d(\eta) = 1.0$ | $log_d(\eta) = 1.5$ | $log_d(\eta) = 2.0$ |
> |:---:|:---:|:---:|:---:|:---:|:---:|
> | $\rho = 0.0$ (Transformer) | **0.4193** | **0.4250** | **0.4214** | **0.3992** | **0.3654** |
> | $\rho = 0.0$ (Equivalent Model) | **0.4200** | **0.4214** |  **0.4200** | **0.3988** | **0.3698** |
> | $\rho = 0.2$ (Transformer) | 0.4263 | 0.4232 | 0.4227 | 0.3969 | 0.3642 |
> | $\rho = 0.2$ (Equivalent Model) | 0.4221 | 0.4200 | 0.4191 | 0.3968 | 0.3761 |
> | $\rho = 0.4$ (Transformer) | 0.4308 | 0.4259 | 0.4276 | 0.3997 | 0.3786 |
> | $\rho = 0.4$ (Equivalent Model) | 0.4324 | 0.4261 | 0.4280 | 0.3975 | 0.3864 |
> | $\rho = 0.6$ (Transformer) | 0.4297 | 0.4379 | 0.4344 | 0.4176 | 0.3814 |
> | $\rho = 0.6$ (Equivalent Model) | 0.4293 | 0.4387 | 0.4319 | 0.4173 | 0.3894 |
> | $\rho = 0.8$ (Transformer) | 0.4552 | 0.4493 | 0.4553 | 0.4246 | 0.3931 |
> | $\rho = 0.8$ (Equivalent Model) | 0.4555 | 0.4476 | 0.4534 | 0.4232 | 0.4023 |
> | $\rho = 1.0$ (Transformer) | 0.4794 | 0.4820 | 0.4754 | 0.4437 | 0.4101 |
> | $\rho = 1.0$ (Equivalent Model) | 0.4757 | 0.4827 | 0.4727 | 0.4458 | 0.4205 |
>
>
> **Key Observations:**
> - In line with our theoretical predictions, the ICL errors of the Transformer model closely match those of the equivalent model, even in this real-world multilingual setting.
> - We observe a clear trend (similar to Fig. 3b in the paper): increasing the GD step size leads to reduced ICL error, indicating effective feature learning.
> - Performance improves as the ratio of English reviews increases, which aligns with the known strength of the embedding model on English data.
>
>
> **Experimental Details**
> - Labels: The review star ratings are demeaned and scaled to lie in the range $y_i \in [-1, 1]$, making the task regression-like.
> - Inputs: Each review text is embedded using the multilingual text embedding model multilingual-e5-small (Wang et al., Microsoft 2024). These 384-dimensional embeddings are reduced to 64 dimensions via PCA, and then normalized. This pre-processing converts each review into a vector input $\mathbf{x}_i$ compatible with our ICL setting.
>
> After this, we form samples for the ICL task following our standard setup.
>
> - Experiment parameters: $d = l = 64$, $n = k = 0.25d^2$, $\lambda = 5 \times 10^{-5}$
>
> - The rest of the details match those of Fig. 3b in the paper.
>
> ---
>
> Thank you for the feedback that motivated this new and interesting experiment. We hope these new results help address your concern about the connection between our findings and real-world tasks. Please let us know if you have any further questions or concerns.

---

### Official Review · Reviewer_JzvP · 2025-07-08

**Clarity:** 2
**Significance:** 3
**Originality:** 2
**Rating:** 5
**Confidence:** 3

**Summary:**

This research paper investigates in-context learning (ICL) in pre-trained transformer models with non-linear multilayer perceptron (MLP) layers, focusing on non-linear regression tasks derived from diverse datasets with heterogeneous input, task, and noise distributions. It builds upon previous efforts that analyzed ICL dynamics in the limited context of linear regression using attention-only models. Leveraging prior studies on feature learning in two-layer neural networks following a single gradient step—particularly in structured Gaussian mixture settings—this work extends those insights to transformer architectures incorporating non-linear MLP layers. Additionally, it examines the interplay between data mixing and feature learning, demonstrating the significance of structured task distributions in facilitating effective feature representation. The efficacy of ICL is assessed using a uniform mixture of data sources. Empirical simulations reveal that the performance of non-linear MLP-based transformers closely approximates the Hermite polynomial surrogate, even in moderate-dimensional settings. Further analysis explores the impact of mixing structured and unstructured data on overall model performance.

**Questions:**

1) Prior work has shown that linear transformers exhibit gradient descent-like behavior during in-context learning. Does the equivalence established in this study imply a similar mechanism in the linear setting? This will help connecting the results prior efforts to explain ICL.
2) For multi-layer transformers with non-linear MLPs, does the theoretical connection to Gaussian Universality continue to hold?  Any simplifying assumptions to explain ICL for multi-layer transformer models with MLP?
3)  Can the analytical techniques used here be extended to explain ICL behavior in models with multiple layers of bi-directional LSTM or xLSTM architectures?

**Ethical Concerns:**

["NO or VERY MINOR ethics concerns only"]

**Final Justification:**

I carefully reviewed the authors’ responses to my comments. My observations regarding their rebuttal are as follows:
- The authors acknowledged the validity of the concerns raised and responded in general terms. However, the responses did not offer any substantive analysis beyond what was already presented in the original manuscript.
Regarding the authors’ responses to comments from other reviewers, I note the following:
- On real-world applicability: The authors present a comparison using a simple sentiment analysis task. While the results demonstrate a reasonable alignment between their model and a standard Transformer, it is important to recognize that this constitutes a toy example and may not reflect broader applicability.
- On the discussion of linear Transformer models and two-layer MLPs with frozen layers: The authors have made an effort to incorporate and contrast relevant literature that was previously omitted. It is worth emphasizing that this paper represents one of the early attempts to explain in-context learning (ICL) using simplified model architectures.
Overall, the paper introduces tools and methodologies that contribute to understanding the learning dynamics of ICL. These techniques may serve as a foundation for future work. Based on these observations, I will maintain my current score.

**Limitations:**

Yes

**Paper Formatting Concerns:**

The paper is formatted according to the guidelines.

**Quality:**

3

**Strengths And Weaknesses:**

Strengths:
1) Provides a detailed analysis of the learning dynamics of in-context learning (ICL) in transformer models with non-linear MLP layers, applied to non-linear regression tasks.
2) Establishes the asymptotic equivalence between transformers with non-linear MLP layers and Hermite polynomial representations, in terms of ICL error—offering a tractable analytical model for studying such architectures.
3) Supports the theoretical framework with empirical evaluations, demonstrating close alignment between the analytical model and simulation results even at moderate dimensions.
4) Empirical results further indicate that the quality of data sources—specifically, those with structured covariance—plays a significant role in reducing ICL error
5) The appendix also includes a useful extension to the data sources having non-zero mean.

Weaknesses:
1) Prior work has shown that linear transformers exhibit gradient descent-like behavior during in-context learning. Does the equivalence established in this study imply a similar mechanism in the linear setting?
2) For multi-layer transformers with non-linear MLPs, does the theoretical connection to Gaussian Universality continue to hold?  Any simplifying assumptions to explain ICL for multi-layer transformer models with MLP?
3)  Can the analytical techniques used here be extended to explain ICL behavior in models with multiple layers of bi-directional LSTM or xLSTM architectures?

---

> ### Author Rebuttal · Authors · 2025-07-29
>
> Thank you for your thoughtful and constructive review. We are pleased that you recognized key strengths of our work, including the detailed analysis of in-context learning (ICL) with Transformers involving nonlinear MLPs, the asymptotic equivalence to Hermite polynomial representations, and the strong alignment between theoretical and empirical findings. Below, we address each of your insightful questions:
>
> > 1. Prior work has shown that linear transformers exhibit gradient descent-like behavior during in-context learning. Does the equivalence established in this study imply a similar mechanism in the linear setting?
>
> Yes, our results are fully compatible with prior work on linear Transformers. Specifically, in the linear case, the analysis reduces to the known gradient descent-like behavior for ICL. The novel contribution here lies in extending this understanding to Transformers with nonlinear MLP heads. We show that such architectures yield predictors equivalent to finite-degree Hermite polynomial surrogates—thereby enriching the class of functions learnable in-context. This extension provides a principled explanation for improved ICL performance on nonlinear tasks, particularly in mixed or structured data scenarios.
>
> > 2. For multi-layer transformers with non-linear MLPs, does the theoretical connection to Gaussian Universality continue to hold? Any simplifying assumptions to explain ICL for multi-layer transformer models with MLP?
>
> This is an excellent question and a natural direction for future research. While the extension to multi-layer Transformers is conceptually plausible, our current analysis relies on the simplification afforded by a single gradient-like step (for the first layer of the MLP). This enables precise derivation of asymptotic equivalence results under Gaussian universality. Multi-layer architectures introduce additional complexity due to inter-layer interactions and compositional nonlinearities. Extending our analysis to this setting would likely require additional assumptions—e.g., layer-wise independence or structure—to preserve analytical tractability.
>
> > 3. Can the analytical techniques used here be extended to explain ICL behavior in models with multiple layers of bi-directional LSTM or xLSTM architectures?
>
> Extending our techniques to recurrent architectures such as bi-directional LSTMs or xLSTMs is non-trivial, primarily due to their architectural differences—especially temporal recurrence and gating mechanisms. Nonetheless, the underlying analytical principles in our work, including Gaussian universality and polynomial surrogate modeling, offer a potential foundation for exploring such extensions. Achieving rigorous theoretical insights in these settings would, however, require new developments to handle the temporal and stateful dynamics unique to recurrent models.
>
> ---
>
> Overall, we sincerely appreciate your positive assessment and the thought-provoking questions, which we believe point toward exciting future research directions.

---

### Comment · Area_Chair_PUqn · 2025-08-05
**Please Participate in the Author-Reviewer Discussion**

Dear Reviewers,

Thank you again for your commitment and engagement in the review process. If you have not done so, we kindly ask that you take the time to read the other reviews and author responses thoroughly. Note that, in addition to the mandatory acknowledgment, please also provide concrete responses to the authors. Your active participation in the Author–Reviewer Discussions is highly valued and helps foster a constructive and productive exchange.

Thank you!

Best regards, AC

---

### Decision · Program_Chairs · 2025-09-17

**Decision:**

Accept (poster)

**Comment:**

This work develops an asymptotic analysis of in-context learning (ICL) in pre-trained transformer models with non-linear MLP heads. The authors present insights into the ICL behaviors of such transformers as well as how the mixture ratio of the heterogeneous pretraining data source will impact the behavior.

While concerns were raised regarding the assumptions on the linear attention, the setup of the gradient descent steps, and the difference from the existing works, the authors did an excellent job in rebuttal in addressing the reviewers' concerns. The new experiments on sentiment classification with transformers provide evidential justification for the alignment between the analyzed setup and the real-world case. By incorporating all the promised revisions in the rebuttal and the discussion period, despite the remaining concerns, the results can be valuable for the community.